# Beyond Logits: Coherent Hallucination Mitigation via Attention Contrastive Decoding

Yujia Chen [1]   Rui Sun [2]   Huayu Mai [1]   Wangkai Li [1]   Zhangyu He [1]   Bingzhou Wang [1]   Aibing Li [1]
Wenzhang SUN [3]   Tianzhu Zhang [1]

## Abstract

Large Vision-Language Models (LVLMs) demonstrate impressive multimodal capabilities, yet suffer from hallucination—generating factually inaccurate content. Contrastive Decoding (CD) mitigates this by contrasting amateur and expert branches at the logit level. However, our investigation reveals that such logit-level interventions fundamentally compromise generation coherence, necessitating restrictive penalty constraints unrelated to hallucination suppression. We introduce Attention Contrastive Decoding (ACD), a training-free plug-in that complements logit-level CD by relocating part of the contrastive operations to the attention mechanism. Operating at an earlier stage of the forward pass, ACD performs smooth semantic-preserving interventions through an Adaptive Subtraction Strategy (ASS), which attenuates hallucination-associated attention patterns while amplifying critical visual information. Extensive experiments demonstrate that combining ACD with existing CD methods (*e.g.,* VCD+ACD) produces substantially more coherent outputs with further reduced hallucinations, eliminating restrictive penalties while enabling trustworthy multimodal generation. Code is available in https://github.com/jj-ccc/ACD.

## 1. Introduction

Large Vision-Language Models (LVLMs) (Liu et al., 2023b;c; 2024a; Zhu et al., 2023; Bai et al., 2023; Li et al., 2023a; Driess et al., 2023; Achiam et al., 2023; Chowdhery et al., 2023; Alayrac et al., 2022) have emerged as a transformative paradigm in multimodal AI. By integrating visual encoders with large language models, LVLMs enable sophisticated cross-modal reasoning across diverse tasks, including image captioning (Lin et al., 2014; Li et al., 2023a), visual question answering (Liu et al., 2023c;b; 2024a; Hudson & Manning, 2019), and complex multimodal reasoning (Lu et al., 2022; Alayrac et al., 2022; Achiam et al., 2023). This advancement establishes a foundation for more natural and intuitive human-machine interaction.

Despite significant advances in LVLMs, the "hallucination" phenomenon remains a critical challenge (Li et al., 2023c; Fu et al., 2023b; Yue et al., 2024). These hallucinations manifest as grammatically coherent outputs with factual inconsistencies relative to visual input. Unlike text-only LLM hallucinations, LVLM hallucinations exhibit distinct cross-modal characteristics, manifesting in three patterns: (1) *Modal dissociation*: generation detached from visual input; (2) *Perceptual fabrication*: hallucination of non-existent attributes (*e.g.,* colors, quantities, spatial relations); (3) *Semantic distortion*: inaccurate visual content summarization. These cross-modal inconsistencies substantially undermine model credibility and pose significant safety risks in critical domains such as medical diagnostics and autonomous driving.

Alleviating hallucinations in LVLMs has emerged as a critical research focus, with training-free Contrastive Decoding (CD) strategies (Li et al., 2025; Huo et al., 2024; Liu et al., 2024c; An et al., 2025; Leng et al., 2024; Wang et al., 2024c; Chen et al., 2024b) representing a prominent paradigm owing to their efficiency, flexibility, and strong interpretability. These approaches mitigate hallucinations through two principal mechanisms: hallucination induction and probability recalibration. For hallucination induction, existing methods can be categorized into three primary classes: (1) Visual Contrastive Decoding (VCD) (Leng et al., 2024) amplifies the model's reliance on linguistic priors by introducing Gaussian noise into visual inputs or completely eliminating visual signals (Liu et al., 2024c), thereby constructing a hallucination-rich probability distribution. (2) Instructional

[1]School of Information Science and Technology / National Key Laboratory of Deep Space Exploration, University of Science and Technology of China, China [2]Shenzhen International Graduate School, Tsinghua University, China [3]School of Software, Tsinghua University, China. Correspondence to: Tianzhu Zhang <tzzhang@ustc.edu.cn>.

*Proceedings of the 43rd International Conference on Machine Learning*, Seoul, South Korea. PMLR 306, 2026. Copyright 2026 by the author(s).

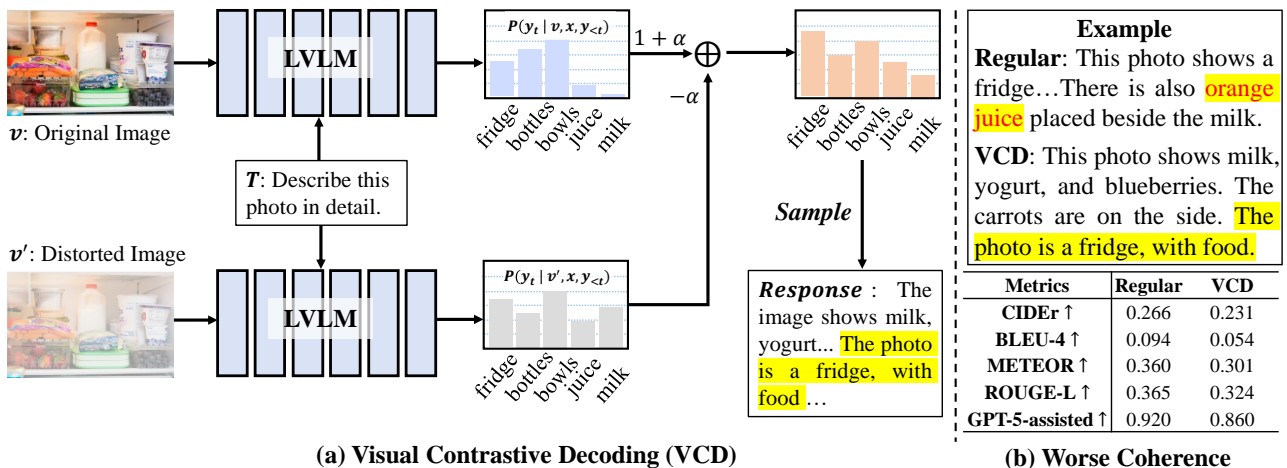

**(a) Visual Contrastive Decoding (VCD)**

**(b) Worse Coherence**

| | Example |
|---|---|
| **Regular**: This photo shows a fridge…There is also ==orange juice== placed beside the milk. |
| **VCD**: This photo shows milk, yogurt, and blueberries. The carrots are on the side. ==The photo is a fridge, with food.== |

| Metrics | Regular | VCD |
|---|---|---|
| **CIDEr** ↑ | 0.266 | 0.231 |
| **BLEU-4** ↑ | 0.094 | 0.054 |
| **METEOR** ↑ | 0.360 | 0.301 |
| **ROUGE-L** ↑ | 0.365 | 0.324 |
| **GPT-5-assisted** ↑ | 0.920 | 0.860 |

*Figure 1.* (a) VCD mitigates hallucinations by subtracting hallucination-induced distributions from original outputs. (b) However, logit-level subtraction disrupts linguistic coherence compared to vanilla decoding .

Contrastive Decoding (ICD) (Wang et al., 2024c) employs adversarial prompts (*e.g.,* "you are a confused object detector") to elicit hallucinated outputs. (3) Self-Introspective Decoding (SID) (Huo et al., 2024) strategically masks high-attention visual regions during decoding to provoke hallucination generation. The probability recalibration mechanism then subtracts this artificially constructed hallucination distribution from the original generation probability. As illustrated in Figure 1(a), this approach effectively suppresses factually inconsistent content while enhancing the model's fidelity to visual evidence, offering a convenient and theoretically interpretable solution for hallucination mitigation in LVLMs.

Despite substantial empirical evidence (Leng et al., 2024; Wang et al., 2024c; Huo et al., 2024) demonstrating the efficacy of contrastive decoding (CD) strategies in mitigating LVLM hallucinations, our investigation reveals that CD approaches significantly compromise generation coherence—manifested as invalid token outputs, abrupt semantic transitions, and structural degradation—under specific operational conditions. The fundamental limitation stems from CD's crude direct subtraction operation at the logits layer, which lacks nuanced regulation of the natural language generation process. Mathematically, this operation is equivalent to re-weighting the probability space via division, *i.e.,* $P_{\text{final}} \propto P_{\text{original}}/P_{\text{distorted}}^{\alpha}$. While theoretically sound for suppressing hallucinations, such aggressive penalization induces a critical side effect known as the "false positive" problem (Li et al., 2023b). Since the distorted model (negative view) retains its fundamental language modeling capabilities, it legitimately assigns high probabilities to high-frequency functional tokens (*e.g.,* "the", "of"). Consequently, direct logit subtraction indiscriminately penalizes these syntactically essential tokens, causing the decoding

trajectory to deviate from the natural language manifold (Liu et al., 2021).

Furthermore, due to the inherent discontinuity and volatility of logits distributions, logit-level subtraction forcibly distorts vocabulary distribution probabilities, thereby disrupting the model's intrinsic language generation mechanisms. This direct intervention inadequately accounts for contextual dependencies and logical coherence, resulting in semantic fragmentation and structural collapse in generated outputs, as visualized in Figure 1(b). Notably, since ICD and SID also operate through logit-level subtraction, they inevitably encounter identical limitations. Our systematic experiments with LLaVA-1.5-7B (Liu et al., 2023b) on the CHAIR benchmark (Yue et al., 2024) corroborate these observations: VCD consistently underperforms across multiple evaluation metrics, including CIDEr (description quality), METEOR (semantic comprehension), ROUGE-L (information completeness), and BLEU (generation precision). Additionally, GPT-5-assisted evaluation scores significantly trail those of conventional decoding methods. These empirical findings explicitly expose an inherent contradiction within current contrastive decoding paradigms—while logit-level contrastive operations effectively suppress hallucinations, they inevitably degrade linguistic quality, presenting a critical trade-off challenge for practical LVLM deployment.

Concurrently, probability recalibration mechanisms at the logits level frequently amplify tail token probabilities due to distribution discontinuities, resulting in invalid token generations. To mitigate this issue, existing contrastive decoding strategies typically incorporate adaptive penalty mechanisms to filter low-probability tokens. However, our in-depth analysis reveals that such compensatory mechanisms not only fail to adequately resolve the underlying issue but

also introduce additional challenges. **First, penalty mechanisms that mask low-probability tokens sacrifice decoding diversity** (constraining the model to high-probability tokens exclusively). This implies that penalty mechanisms function primarily to counteract CD's adverse effects rather than addressing the hallucination problem directly. This compensatory processing significantly restricts the richness and creativity of generated content, biasing outputs toward high-frequency, conventional expressions lacking natural linguistic variation and innovation. **Second, the intensity calibration of current penalty mechanisms predominantly relies on heuristic algorithms** (utilizing manually defined thresholds), severely limiting their adaptability and inevitably resulting in invalid token outputs. In diverse linguistic contexts and tasks, fixed penalty strategies inadequately accommodate variable generation requirements and cannot dynamically adjust filtering intensity based on contextual semantics, further exacerbating the inconsistency and instability of generated outputs.

To fundamentally address these limitations prevalent in existing CD methods, we propose Attention Contrastive Decoding (ACD), which relocates contrastive operations from the discrete logit space to the continuous attention layer. This architectural choice leverages the attention mechanism's inherent capacity to weight input information and adapt to contextual cues, facilitating smooth information flow (Vaswani et al., 2017; Castin et al., 2023). Consequently, ACD naturally generates more coherent outputs while circumventing the need for penalty terms and preserving generation diversity (Detailed theoretical and empirical analyses are provided in the Appendix B and C). This adaptive contrastive mechanism at the attention level not only precisely localizes and suppresses hallucination sources but also preserves the model's intrinsic language generation capabilities, offering a more elegant and effective solution for hallucination mitigation in LVLMs. Beyond architectural relocation, ACD introduces an Adaptive Subtraction Strategy (ASS) for finer-grained hallucination control. Unlike uniform subtraction in traditional CD, ASS distinguishes between two fundamentally different attention shift patterns under hallucination induction:(1) **information degradation**, where decreased attention signals loss of focus on semantically relevant content that should be preserved, and (2) **hallucination risk**, where increased attention indicates over-focus on spurious regions that warrant suppression. Based on this assessment, ASS adaptively modulates the model's attention allocation across different regions. Specifically, it amplifies attention to areas exhibiting information degradation while suppressing regions with hallucination risk, thereby inhibiting hallucination generation at its source and enhancing focus on semantically valid content. This fine-grained control mechanism enables the model to maintain linguistic quality while improving effective information utilization.

In summary, our main contributions are three-fold: (1) We reveal that logit-level subtraction in CD fundamentally compromises generation quality, and that penalty mechanisms fail to resolve invalid token generation while restricting output diversity. (2) We propose Attention Contrastive Decoding (ACD), which relocates contrastive operations to the attention layer, leveraging attention's inherent continuity to preserve coherence without penalties. We further introduce the Adaptive Subtraction Strategy (ASS), which distinguishes information degradation from hallucination risk to achieve precise, selective suppression. (3)Extensive experiments demonstrates ACD's capacity to generate more coherent, higher-quality content with significantly reduced hallucinations without penalty mechanisms, advancing the SOTA in reliable LVLMs decoding.

## 2. Related Work

### 2.1. Hallucination in LVLMs

Research addressing hallucinations in large vision-language models (LVLMs) can be categorized into three primary approaches. *First, contrastive decoding techniques* identify and suppress hallucinated content without parameter updates. Representative methods include VCD (Leng et al., 2024), ICD (Wang et al., 2024c), and SID (Huo et al., 2024), with advanced variants such as HALC (Chen et al., 2024b), VaLiD (Wang et al., 2024b), CMVED (Li et al., 2025), PAI (Liu et al., 2024c), and AGLA (An et al., 2025) further refining these techniques through targeted attention manipulation. *Second, fine-tuning strategies* employ curated datasets incorporating negative (Liu et al., 2023a), counterfactual (Yu et al., 2024), or purified data (Wang et al., 2024c). Related frameworks include HALVA (Sarkar et al., 2024), HACL (Jiang et al., 2024a), PerturboLLaVA (Chen et al., 2025a), PATCH (Shang et al., 2024), HIO (Lyu et al., 2024), Octopus (Suo et al., 2025), OPA (Yang et al., 2025b), TL-DPO (Yoon et al., 2025), and VASparse (Zhuang et al., 2025). *Third, direct intervention methods* manipulate visual features or attention mechanisms. These include DeCo (Wang et al., 2024a), VTI (Liu et al., 2024b), VDGD (Ghosh et al., 2024), ProjectAway (Jiang et al., 2024b), TAME (Tang et al., 2025a), OPERA (Huang et al., 2024), VAR (Jiang et al., 2025b), FarSight (Tang et al., 2025b), Nullu (Yang et al., 2025a), and ClearSight (Yin et al., 2025). Notably, our method migrates contrastive decoding from logit level to attention layer, leveraging internal consistency for self-calibration rather than directly intervening in attention mechanisms.

### 2.2. Contrastive Decoding in Hallucination

Contrastive Decoding (CD) methods provide an elegant, training-free approach to mitigating hallucinations in LVLMs without parameter optimization. These strategies

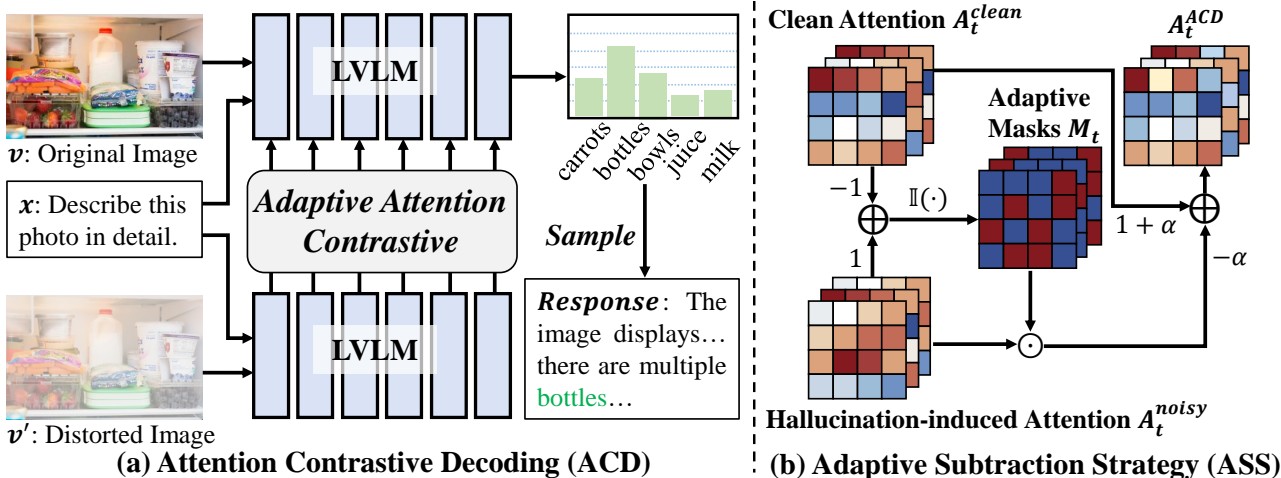

**(a) Attention Contrastive Decoding (ACD)**   **(b) Adaptive Subtraction Strategy (ASS)**

*Figure 2.* **Overview of the proposed framework (VCD + ACD). (a) Attention Contrastive Decoding (ACD).** Given an image $v$ and its perturbed counterpart $v'$, ACD computes parallel attention maps $A_t^{\text{clean}}$ and $A_t^{\text{noisy}}$ within the decoder. Contrastive subtraction is performed in the *continuous attention space* (Equation 9), producing a refined attention distribution $A_t^{\text{ACD}}$ that sharpens focus on visually grounded regions while suppressing hallucination-prone activations. The resulting logits are produced by the refined attention and decoded directly, *replacing* the logit-level subtraction and penalty mechanism of conventional CD methods. **(b) Adaptive Subtraction Strategy (ASS).** Rather than applying uniform subtraction, ASS constructs a binary mask $M_t$ that distinguishes *signal-degraded* positions (amplified by $(1+\alpha)$) from *hallucination-risk* positions (suppressed via contrastive subtraction), yielding a modulated attention map with an enhanced signal-to-noise ratio.

reduce hallucination generation by contrastively adjusting token probability distributions through two key mechanisms: hallucination induction and probability adjustment. Representative works include VCD (Leng et al., 2024), which injects Gaussian noise into visual inputs to construct hallucination-rich distributions; ICD (Wang et al., 2024c), which employs negative prompts to induce hallucinations; and SID (Huo et al., 2024), which strategically masks high-attention regions to expose model vulnerabilities. Advanced variants include HALC (Chen et al., 2024b) with automatic hallucination correction, VaLiD (Wang et al., 2024b) comparing early-layer outputs, CMVED (Li et al., 2025) masking cross-modal attention, PAI (Liu et al., 2024c) removing visual components entirely, and AGLA (An et al., 2025) emphasizing critical regions through image-text matching models. Several recent works also leverage attention mechanisms for hallucination mitigation, yet differ fundamentally from ACD. FarSight (Tang et al., 2025b) and DAMRO (Gong et al., 2024) modify attention within a single forward pass or use attention to select distorted tokens, but perform contrastive subtraction at the *logit level*. HICD (Jiang et al., 2025a) disperses attention to improve hallucination induction, again feeding results into logit-level subtraction. MoD (Chen et al., 2025c) adaptively switches between addition and subtraction but operates entirely on logits. In contrast, ACD is the first to relocate the contrastive subtraction itself from the logit space to the attention layer, eliminating the coherence degradation and penalty mechanisms inherent in all logit-level approaches. Despite demonstrated efficacy in hallucination mitigation, conventional CD

strategies operate exclusively at the logits distribution level, substantially compromising output coherence and diminishing generative diversity.

## 3. Method

### 3.1. Preliminary

**Vanilla Decoding.** A standard LVLM $\theta$ comprises a vision encoder, a vision-text interface, and an LLM decoder. Given an image $v$ and text query $x$, the model autoregressively generates output $y$ via:

$$y_t \sim p_\theta(y_t|v, x, y_{<t}) \propto \exp(\text{logit}_\theta(y_t|v, x, y_{<t})), \quad (1)$$

where $y_t$ is the $t$-th generated token, $y_{<t}$ denotes all preceding tokens, and $\text{logit}_\theta$ returns the pre-softmax scores. The attention weight matrix within each decoder head is:

$$A = \text{Softmax}\left(\frac{QK^\top}{\sqrt{d}}\right), \quad (2)$$

where $Q, K \in \mathbb{R}^{L \times d}$ are the query and key matrices, $L$ is the sequence length, and $d$ is the head dimension. The output is then $AV$ with value matrix $V \in \mathbb{R}^{L \times d}$.

During autoregressive inference, the model caches $K$ and $V$ from all previous steps. The attention at step $t$ reduces to:

$$a_t = \text{Softmax}\left(\frac{q_t K_{\leq t}^\top}{\sqrt{d}}\right), \quad (3)$$

where $q_t \in \mathbb{R}^{1 \times d}$ is the current query and $K_{\leq t}$ aggregates cached keys. This cached attention computation serves as the intervention point for our proposed ACD.

**Contrastive Decoding.** CD (Leng et al., 2024; Wang et al., 2024c; Huo et al., 2024) mitigates hallucinations by subtracting a hallucination-rich distribution from the original output. The hallucination distribution is induced by perturbing inputs (visual corruption $v'$ (Leng et al., 2024), adversarial prompts $x'$ (Wang et al., 2024c), or attention masking (Huo et al., 2024)). Subsequently, a new contrastive probability distribution is computed by leveraging the differences between these two distributions. This contrastive distribution, denoted as $p_{cd}$, is defined as:

$$p_{cd}\left(y \mid v, v', x'\right) = \text{softmax}\left[(1 + \alpha)\,\text{logit}_\theta\left(y \mid v, x\right)\right.$$
$$\left. -\alpha\,\text{logit}_\theta\left(y \mid v', x'\right)\right], \quad (4)$$

where larger $\alpha$ values indicate a stronger amplification of differences between the two distributions ($\alpha = 0$ degenerates to Vanilla decoding). A fundamental limitation of Equation 4 is that the CD subtraction operation can drive certain reasonably token logits to extremely small or negative values, causing unreasonably low-probability tokens to emerge after softmax normalization and thus amplifying tail token probabilities. This necessitates a uniform penalization constraint that suppresses all low-probability outputs from the original inputs:

$$\mathcal{V}_{\text{head}}\left(y_{<t}\right) = \{y_t \in \mathcal{V} :$$
$$p_\theta\left(y_t \mid v, x, y_{<t}\right) \geq \beta \max_w p_\theta\left(w \mid v, x, y_{<t}\right)\}, \quad (5)$$
$$p_{cd}\left(y_t \mid v, v', x'\right) = 0, \, if\, y_t \notin \mathcal{V}_{\text{head}}\left(y_{<t}\right),$$

where $\mathcal{V}$ is the output vocabulary of LVLMs and $\beta$ is a hyperparameter in $[0, 1]$ for controlling the truncation of the next token distribution. Larger $\beta$ indicates more aggressive truncation, keeping only high-probability tokens. Combining the CD and the plausibility constraint, we obtain the full formulation:

$$y_t \sim \text{softmax}\left[(1 + \alpha)\,\text{logit}_\theta\left(y_t \mid v, x, y_{<t}\right)\right.$$
$$\left. -\alpha\,\text{logit}_\theta\left(y_t \mid v', x', y_{<t}\right)\right], \, subject\, to\, y_t \in \mathcal{V}_{\text{head}}\left(y_{<t}\right). \quad (6)$$

### 3.2. Attention Contrastive Decoding (ACD)

Existing CD methods (Equation 6) perform contrastive subtraction in the discrete logit space, where even small perturbations can cause abrupt redistributions over the vocabulary, degrading linguistic coherence and necessitating ad-hoc penalty terms (Equation 5). We propose **Attention Contrastive Decoding (ACD)**, a lightweight plug-in that introduces contrastive operations into the *continuous attention layer*, complementing logit-level CD to jointly suppress hallucinations. The complete framework is illustrated in Figure 2.

**Motivation.** Attention weights reside in a continuous probability simplex produced by the Softmax operator, where arithmetic combinations yield semantically smooth transitions (Vaswani et al., 2017; Castin et al., 2023). In contrast, logit subtraction operates on unbounded pre-softmax scores whose exponentiation amplifies minor differences into drastic probability shifts. By intervening before the

logit stage, ACD modulates *where* the model attends rather than *what* it outputs, inherently preserving the linguistic structure of the downstream distribution.

**Formulation.** At decoding step $t$, we compute attention weights under two input configurations—the original input $(v, x)$ and a hallucination-inducing input $(v', x')$:

$$A_t^{\text{clean}} = \text{Softmax}\left(\frac{q_t\, K_{\leq t}^{\text{clean}\top}}{\sqrt{d}}\right),$$
$$A_t^{\text{noisy}} = \text{Softmax}\left(\frac{q_t\, K_{\leq t}^{\text{noisy}\top}}{\sqrt{d}}\right), \quad (7)$$

where $q_t \in \mathbb{R}^{1 \times d}$ is the query at step $t$, and $K_{\leq t}^{\text{clean}}$, $K_{\leq t}^{\text{noisy}} \in \mathbb{R}^{t \times d}$ are cached key matrices under the respective inputs. Unlike logit-level CD (Equation 6), ACD performs contrastive subtraction directly on attention weights:

$$A_t^{\text{base}} = (1 + \alpha)\, A_t^{\text{clean}} - \alpha\, A_t^{\text{noisy}}, \quad (8)$$

where $\alpha > 0$ controls the contrastive strength. This operation amplifies attention on regions consistently highlighted by the clean model while suppressing those emphasized only under noise. For generality, we apply ACD uniformly across all decoder layers and attention heads.

**Compatibility with Existing CD.** ACD serves as a modular alternative to the logit-level subtraction and penalty mechanism shared by existing CD methods. Given any hallucination induction strategy (*e.g.,* noise injection in VCD, adversarial prompts in ICD, attention masking in SID), ACD performs the contrastive operation at the attention layer instead of the logit layer, thereby inheriting each method's hallucination-inducing capability while avoiding the coherence degradation and auxiliary penalty terms inherent in logit-level subtraction.

### 3.3. Adaptive Subtraction Strategy (ASS)

Equation 8 applies uniform subtraction across all positions, implicitly assuming that noise universally inflates attention on irrelevant regions. In practice, however, the effect of noise is *bidirectional*. We analyze attention shifts between $A_t^{\text{clean}}$ and $A_t^{\text{noisy}}$ over 500 COCO image-caption pairs and observe two distinct phenomena: **(i)** approximately 32% of positions exhibit *attention degradation* ($A_t^{\text{clean}} > A_t^{\text{noisy}}$), where noise obscures valid visual features; **(ii)** the remaining 68% show *attention inflation* ($A_t^{\text{clean}} \leq A_t^{\text{noisy}}$), indicating hallucination risk. Applying uniform subtraction to group (i) would counter-productively penalize ground-truth regions. Motivated by this finding, we propose the **Adaptive Subtraction Strategy (ASS)**, which applies position-dependent operations:

- **Case 1 — Signal Amplification** ($A_t^{\text{clean}}[i] > A_t^{\text{noisy}}[i]$). Noise has degraded attention on semantically valid

Question: Is there a person in the image?

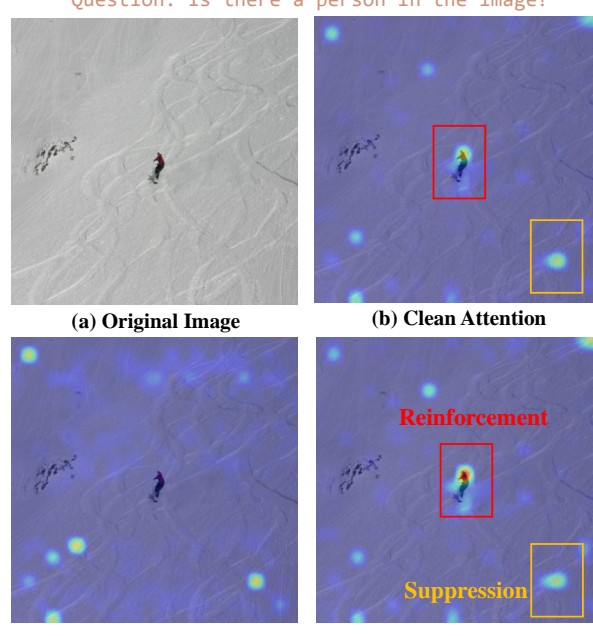

**(a) Original Image**                **(b) Clean Attention**

**(c) Hallucination-induced Attention**    **(d) Attention for ASS**

*Figure 3.* **Visualization of the Adaptive Subtraction Strategy (ASS).** (a) Input image. (b) Clean attention correctly focuses on the person. (c) Under hallucination induction, attention drifts to background noise. (d) **Attention after ASS.** The red box marks a Case 1 region ($A^{\mathrm{clean}} > A^{\mathrm{noisy}}$): ASS amplifies the clean signal by $(1+\alpha)$, producing sharper focus than (b). The yellow box marks a Case 2 region ($A^{\mathrm{noisy}} \geq A^{\mathrm{clean}}$): ASS suppresses spurious activation via contrastive subtraction. The resulting map (d) achieves a higher signal-to-noise ratio than the original clean state (b), widening the margin between object and background.

regions. Rather than subtracting, we *amplify* these positions to reinforce correct visual grounding.

- **Case 2 — Hallucination Suppression** ($A_t^{\mathrm{clean}}[i] \leq A_t^{\mathrm{noisy}}[i]$). Elevated attention signals attraction to spurious correlations. We apply contrastive *suppression* to attenuate these high-risk positions.

**Formulation.** We define a binary hallucination-risk mask $M_t = \mathbb{I}\Big(A_t^{\mathrm{noisy}} > A_t^{\mathrm{clean}}\Big) \in \{0,1\}^L$ and compute the modulated attention as:

$$A_t^{\mathrm{ACD}} = \underbrace{(1+\alpha)\,A_t^{\mathrm{clean}}}_{\text{Global Amplification}} - \underbrace{\alpha\Big(A_t^{\mathrm{noisy}} \odot M_t\Big)}_{\text{Selective Suppression}}, \qquad (9)$$

where $\odot$ denotes element-wise multiplication. The formulation elegantly unifies both cases: when $M_t[i] = 0$ (Case 1), the subtraction term vanishes and the clean attention is amplified by $(1+\alpha)$; when $M_t[i] = 1$ (Case 2), contrastive subtraction actively suppresses spurious activations. As visualized in Figure 3, ASS does not merely restore the clean attention map—it actively *expands* the margin between relevant and irrelevant regions, functioning as a bidirectional

signal-to-noise filter that enhances discriminative power beyond the original clean baseline.

## 4. Experiments

### 4.1. Experimental Settings

Following established evaluation protocols from prior work (Leng et al., 2024; Huo et al., 2024; An et al., 2025), we assess our method's efficacy across standard benchmarks. Comprehensive details regarding datasets, evaluation metrics, case studies, and additional ablation experiments are provided in the Appendix H.

**Benchmarks. CHAIR** (Yue et al., 2024): Quantifies hallucination using image annotations rather than lexical similarity. Primary metrics include $CHAIR_s$ (sentence-level), $CHAIR_i$ (instance-level), and recall for measuring semantic comprehensiveness. **LLaVA-Bench (In-the-Wild)** (Liu et al., 2023b): Evaluates generalization through 24 challenging images (memes, paintings, sketches) with 60 questions. Performance assessed by GPT-5 (OpenAI, 2025) on factual accuracy and descriptive richness. **POPE** (Li et al., 2023c): A binary classification framework (20,000+ QA pairs) assessing object hallucination through *yes/no* questions with random, popular, and adversarial sampling strategies. Performance measured via *Accuracy, Precision, Recall,* and *F1*. **MME** (Fu et al., 2023a): Comprehensive evaluation across 14 subtasks spanning perception (object existence, color, position) and cognition (reasoning, computation). Each image features complementary *yes/no* questions evaluated through accuracy metrics.

**Implementation Details.** We evaluate the effectiveness of our proposed ACD on several established LVLMs, including LLaVA-1.5 (Liu et al., 2023b) (7B and 13B) with MLP projection layers and Qwen-VL (7B) (Bai et al., 2023) with q-former projection layers. All experiments are conducted on NVIDIA RTX 3090 GPUs unless otherwise specified. LLaVA-1.5-13B requires an A6000 GPU due to memory constraints. By default, all models refer to 7B variants (*e.g.,* LLaVA-1.5, Qwen-VL) unless explicitly stated. To demonstrate the efficacy of our approach, we compare ACD against established CD strategies designed for hallucination mitigation in LVLMs, including VCD (Leng et al., 2024), ICD (Wang et al., 2024c), and SID (Huo et al., 2024). Furthermore, we integrate ACD into these existing contrastive frameworks to evaluate its compatibility and effectiveness across diverse hallucination-inducing scenarios. Specifically, we denote **X+ACD** as the variant where the original logit-level subtraction and adaptive penalty terms in method X are *entirely replaced* with our attention-level subtraction combined with the ASS strategy; no logit-level contrastive subtraction or penalty filtering (Equation 5) is applied. Thus, all X+ACD results reflect ACD operating as a *standalone* de-

*Table 1.* Results of CHAIR hallucination evaluation for the open-ended caption generation task.

| Model | LLaVA-1.5 | | | Qwen-VL | | |
|---|---|---|---|---|---|---|
| **Decoding** | $Chair_s \downarrow$ | $Chair_i \downarrow$ | $Recall\uparrow$ | $Chair_s \downarrow$ | $Chair_i\downarrow$ | $Recall \uparrow$ |
| Regular | 54.1 | 18.5 | 73.4 | 50.5 | 15.0 | 71.1 |
| VCD | 51.8 | 16.2 | 76.8 | 47.5 | 13.7 | 71.5 |
| **VCD+ACD** | **51.0** | **14.3** | **78.3** | **47.0** | **12.5** | **72.1** |
| ICD | 52.1 | 15.5 | 76.7 | 48.1 | 13.1 | 69.6 |
| **ICD+ACD** | **50.2** | **13.9** | **78.5** | **46.5** | **12.6** | **71.9** |
| SID | 50.5 | 14.1 | 78.2 | 46.0 | 12.7 | 72.5 |
| **SID+ACD** | **49.6** | **13.3** | **79.4** | **45.0** | **11.1** | **73.6** |

coding mechanism under different noise constructions. For VCD+ACD, the integration is straightforward as no special handling is required. For ICD+ACD, we ensure identical prompt lengths between original and hallucination-induced inputs to maintain attention alignment. For SID+ACD, we first mask high-weight patches and apply smoothing, then perform attention-layer subtraction. Note that throughout all experiments, we maintain consistent hyperparameters and employ sampling as the default decoding strategy.

### 4.2. Experimental Results

**Experiments on CHAIR.** Unlike binary response paradigms (*yes/no*) in POPE and MME, the CHAIR dataset requires detailed descriptive generation, increasing hallucination susceptibility. Table 1 compares three representative CD baselines (VCD, ICD, SID) across two architectures (LLaVA-1.5, Qwen-VL), revealing three key findings: **(1) Consistent hallucination reduction.** ACD integration yields substantial improvements across all configurations. For LLaVA-1.5, ACD reduces $Chair_i$ by 11.7% (VCD: 16.2→14.3), 10.3% (ICD: 15.5→13.9), and 5.7% (SID: 14.1→13.3) while improving recall. Qwen-VL shows comparable gains (VCD+ACD: 8.8% reduction to $Chair_i$ 12.5), demonstrating that ACD's generalizability enables seamless integration into CD methods. **(2) Simultaneous improvements in accuracy and coverage.** ACD simultaneously improves both dimensions across all 6 configurations: recall increases by 0.6-2.3 points alongside hallucination reductions. ACD and ASS smoothly suppress attention to hallucination-prone regions while enhancing grounded object attention, enabling detailed yet accurate captions. **(3) Architecture-agnostic gains.** Despite architectural differences, improvements remain consistent: LLaVA-1.5 and Qwen-VL achieve 9.2% and 8.4% average $Chair_i$ reductions with 1.9 and 1.3 point recall gains respectively. This stability indicates ACD exploits fundamental transformer attention properties rather than model-specific artifacts.

**Results on LLaVA-Bench-Wild.** Following established protocols (Leng et al., 2024; Huo et al., 2024; An et al., 2025), we employ GPT-5 (OpenAI, 2025) as an independent assessor to score *Accuracy* (factual correctness) and *Detail* (comprehensiveness) on a 10-point scale. Results are reported in Table 2. Integrating ACD with VCD yields consistent gains across both models. On LLaVA-1.5, VCD+ACD improves total Accuracy by +1.80 (4.30→6.10) and Detail by +0.65 (4.75→5.40). On Qwen-VL, gains are more pronounced: +1.71 Accuracy (5.40→7.11) and +0.96 Detail (4.85→5.81). Among the three categories, *Detail* shows the largest boost: +3.00 Accuracy / +1.66 Detail on LLaVA-1.5, and +2.65 / +2.03 on Qwen-VL. This aligns with our motivation—attention-level modulation sharpens focus on fine-grained regions critical for detailed descriptions. *Conversational* tasks also benefit substantially (+2.16 / +1.86 Accuracy), indicating improved factual grounding in open-ended dialogue. Gains on *Complex* reasoning are more modest, suggesting such tasks rely more on language priors where logit-level CD remains the primary mechanism.

**Results on POPE.** Table 5 reports results on the three POPE splits (Random, Popular, Adversarial) when ACD is integrated into VCD, ICD, and SID across LLaVA-1.5 and Qwen-VL. ACD yields consistent F1 improvements across all six baseline–model combinations. On Qwen-VL, VCD+ACD improves F1 by +1.0 on Random (86.8→87.8), +0.4 on Popular, and +1.0 on Adversarial. On LLaVA-1.5, the same pairing yields +0.5/+1.2/+1.1 across the three splits. ICD+ACD and SID+ACD exhibit similar trends, confirming ACD's role as a model- and method-agnostic plug-in. Two observations merit attention. First, the largest gains concentrate on the harder Popular and Adversarial splits, where hallucination pressure is greatest—precisely the regime where adaptive attention modulation (Section 3.3) is most effective. Second, ACD tends to improve Recall more than Precision (*e.g.,* +1.2 Recall vs. −0.4 Precision for VCD on LLaVA-1.5 Random), suggesting that attention-level reinforcement (Case 1) recovers missed objects without introducing new false positives.

**Results on MME.** We further evaluate ACD on the MME benchmark using LLaVA-1.5. MME covers 14 subtasks spanning perception (*e.g.,* color, spatial position) and cognition (*e.g.,* reasoning, computation) with a binary yes/no protocol, providing a broad testbed for both hallucination

*Table 2.* Results of GPT-5 evaluation on the LLaVA-Bench-Wild dataset.

| Model | Decoding | Conv. | | Detail. | | Complex | | Total | |
|---|---|---|---|---|---|---|---|---|---|
| | | Acc. | Detail | Acc. | Detail | Acc. | Detail | Acc. | Detail |
| LLaVA-1.5 | VCD | 4.67 | 3.67 | 4.50 | 5.67 | 3.88 | 4.88 | 4.30 | 4.75 |
| | **VCD+ACD** | **6.83** | **3.71** | **7.50** | **7.33** | **3.97** | **5.21** | **6.10** | **5.40** |
| Qwen-VL | VCD | 5.61 | 3.91 | 5.55 | 5.53 | 5.13 | 4.98 | 5.40 | 4.85 |
| | **VCD+ACD** | **7.47** | **4.58** | **8.20** | **7.56** | **6.02** | **5.42** | **7.11** | **5.81** |

*Table 3.* Results from Evaluation on the MME Benchmark using LLaVA-1.5.

| Method | Perception | Cognition |
|---|---|---|
| Regular | 1440.0 | 294.6 |
| VCD | 1475.4 | 284.3 |
| **VCD+ACD** | **1497.7** | **298.5** |
| ICD | 1464.2 | 287.2 |
| **ICD+ACD** | **1488.6** | **290.3** |
| SID | 1481.4 | 291.7 |
| **SID+ACD** | **1518.7** | **299.1** |

*Table 4.* Hallucination and coherence evaluation on Qwen3-VL. Upper: CHAIR metrics; Lower: generation quality metrics.

| | Method | Qwen3-VL-4B | | | Qwen3-VL-8B | | |
|---|---|---|---|---|---|---|---|
| | | Regular | VCD | **ACD** | Regular | VCD | **ACD** |
| Hallu. | Chair$_s$↓ | 28.3 | 26.1 | **24.1** | 23.2 | 22.6 | **20.8** |
| | Chair$_i$↓ | 11.4 | 10.7 | **8.8** | 8.4 | 8.4 | **7.3** |
| | Recall↑ | 55.2 | 54.7 | **57.8** | 47.2 | 48.7 | **49.1** |
| Coher. | CIDEr↑ | 0.277 | 0.265 | **0.274** | 0.387 | 0.294 | **0.389** |
| | BLEU-4↑ | 0.121 | 0.120 | **0.122** | 0.158 | 0.129 | **0.160** |
| | GPT-5↑ | 0.945 | 0.917 | **0.945** | 0.958 | 0.939 | **0.961** |

suppression and general capability retention. Results are reported in Table 3. ACD consistently improves both Perception and Cognition scores across all three baselines. VCD+ACD raises Perception by +22.3 (1475.4→1497.7) and Cognition by +14.2 (284.3→298.5). ICD+ACD and SID+ACD follow the same trend, with SID+ACD achieving the highest overall scores (1518.7 / 299.1). Notably, the Cognition gains indicate that attention-level modulation does not sacrifice higher-order reasoning for perceptual fidelity—a common trade-off in logit-level CD methods. This confirms that ACD serves as a general-purpose plug-in that enhances factual grounding while preserving model versatility.

**Generalization to Modern Architectures.** To validate ACD on recent LVLMs, we evaluate on Qwen3-VL (Bai et al., 2025) (2025) at both 4B and 8B scales. Table 4 reports CHAIR hallucination metrics alongside generation quality scores on the same benchmark. ACD consistently reduces hallucination while fully recovering the coherence that logit-level VCD degrades. On Qwen3-VL-4B, ACD lowers Chair$_i$ by 22.8% (11.4→8.8) while improving Recall by 2.6 points; coherence metrics (CIDEr, BLEU-4, GPT-5) return to baseline levels. On Qwen3-VL-8B, we observe that VCD produces 12.8% malformed outputs on POPE and 69.2% on MME (*e.g.,* infinite non-English token loops), a catastrophic failure caused by logit subtraction suppressing EOS probability below the penalty threshold in deeper architectures. ACD exhibits zero malformed outputs, as attention-level intervention preserves the natural token ranking without penalty mechanisms. These results confirm that ACD generalizes to modern architectures and that the coherence degradation problem of logit-level CD becomes *more* severe as models scale deeper.

### 4.3. Ablation Study and Analysis

**Ablation on Components.** Table 6 isolates the contribution of each component in our framework. We compare five configurations: Regular decoding, VCD without/with its penalty term, and ACD without/with ASS. Rows 2–3 show that VCD's penalty mechanism reduces hallucination (Chair$_s$: 53.4→51.8) but introduces a threshold hyperparameter. Row 4 replaces the penalty with ACD$_{base}$—uniform attention subtraction (Eq. 8) without ASS—and achieves comparable Chair$_s$ (52.1) while being threshold-free, confirming that attention-level intervention is inherently smooth. Adding ASS (row 5) brings further gains: Chair$_s$ drops by 1.1 (52.1→51.0), Chair$_i$ by 1.3 (15.6→14.3), and Recall rises by 0.9 (77.4→78.3). These improvements stem from ASS's ability to reinforce informative tokens (Case 1) while selectively suppressing hallucination triggers (Case 2), as visualized in Figure 3. The full VCD+ACD framework achieves the best trade-off, reducing Chair$_s$ by 3.1 and improving Recall by 4.9 over Regular decoding.

*Table 6.* Effectiveness of different components on the CHAIR dataset using LLaVA-1.5.

| Method | Pen. | ASS | Chair$_s$ ↓ | Chair$_i$ ↓ | Recall↑ |
|---|---|---|---|---|---|
| Regular | ✗ | ✗ | 54.1 | 18.5 | 73.4 |
| VCD$_{W/O\,pen.}$ | ✗ | ✗ | 53.4 | 17.6 | 74.9 |
| VCD | ✓ | ✗ | 51.8 | 16.2 | 76.8 |
| **VCD+ACD$_{base}$** | ✗ | ✗ | 52.1 | 15.6 | 77.4 |
| **VCD+ACD** | ✗ | ✓ | **51.0** | **14.3** | **78.3** |

**Causal Validation of ASS.** To directly validate that ASS's two cases correspond to their intended semantic roles, we conduct an attention-object alignment analysis on 500 COCO images. We extract $A_t^{clean}$ and $A_t^{noisy}$ over visual tokens and measure each group's attention mass within ground-truth bounding boxes versus hallucinated regions identified by the CHAIR pipeline. Case 1 positions ($A^{clean} > A^{noisy}$) allocate 74.2% of attention to ground-truth objects and only 12.5% to hallucinated regions, while Case 2 positions show the opposite pattern (18.7% vs. 61.3%). This confirms that ASS's binary mask correctly distinguishes signal degradation from hallucination risk without requiring any tunable threshold.

*Table 5.* Experimental results on the three POPE subsets derived from MSCOCO with LLaVA-1.5 and Qwen-VL.

| Model | LLaVA-1.5 | | | | | | | | | | | |
|---|---|---|---|---|---|---|---|---|---|---|---|---|
| **Setting** | *Random* | | | | *Popular* | | | | *Adversarial* | | | |
| **Decoding** | Acc. | Precision | Recall | F1 | Acc. | Precision | Recall | F1 | Acc. | Precision | Recall | F1 |
| Regular | 83.2 | 91.7 | 73.0 | 81.3 | 81.8 | 88.9 | 72.8 | 80.0 | 78.9 | **83.1** | 72.7 | 77.6 |
| VCD | **87.6** | **92.8** | 81.4 | 86.7 | 82.6 | 87.1 | 80.5 | 83.7 | **77.3** | 73.4 | 86.4 | 79.4 |
| **VCD+ACD** | 86.5 | 92.4 | **82.6** | **87.2** | **83.4** | **88.9** | **81.2** | **84.9** | 77.2 | **74.8** | 87.1 | 80.5 |
| ICD | **87.5** | 87.2 | 80.9 | 83.9 | 83.2 | 83.5 | 79.7 | 81.6 | **79.1** | 72.4 | 86.5 | 78.8 |
| **ICD+ACD** | 87.1 | **87.5** | **82.5** | **84.9** | **83.5** | **84.2** | **80.9** | **82.5** | 78.8 | **73.2** | 86.6 | 79.3 |
| SID | 85.2 | 89.4 | 80.8 | 84.9 | **83.7** | **87.9** | 81.2 | 84.4 | 80.1 | 72.4 | 87.6 | 79.3 |
| **SID+ACD** | **86.9** | **90.0** | **82.2** | **85.9** | 83.6 | 86.9 | **84.3** | **85.6** | **81.7** | **73.6** | **88.8** | **80.5** |
| Model | Qwen-VL | | | | | | | | | | | |
| **Setting** | *Random* | | | | *Popular* | | | | *Adversarial* | | | |
| **Decoding** | Acc. | Precision | Recall | F1 | Acc. | Precision | Recall | F1 | Acc. | Precision | Recall | F1 |
| Regular | 84.4 | **95.4** | 72.5 | 82.4 | 84.1 | **94.3** | 72.6 | 82.0 | 82.2 | **89.9** | 72.6 | 80.3 |
| VCD | 87.5 | 93.5 | 81.0 | 86.8 | 86.8 | **92.0** | 80.6 | 85.9 | 81.9 | 85.5 | 80.4 | 82.9 |
| **VCD+ACD** | **88.6** | **94.6** | **81.9** | **87.8** | **87.1** | 91.4 | **81.8** | **86.3** | **84.2** | 85.8 | **82.0** | **83.9** |
| ICD | 87.2 | 93.0 | **81.5** | 86.9 | 85.0 | 90.5 | **79.8** | 84.8 | 80.9 | 84.0 | 77.3 | 80.5 |
| **ICD+ACD** | **88.0** | **94.3** | 80.8 | **87.0** | **85.5** | **90.8** | 79.5 | **84.8** | **81.6** | **84.3** | **78.5** | **81.3** |
| SID | 86.6 | 92.4 | 79.8 | 85.6 | 84.6 | 88.3 | 77.9 | 82.8 | 79.7 | 82.0 | 75.5 | 78.6 |
| **SID+ACD** | **87.3** | **94.0** | **80.4** | **86.7** | **85.2** | **89.9** | **78.7** | **83.9** | **80.3** | **82.8** | **76.9** | **79.7** |

**Efficiency.** Since ACD replaces logit-level subtraction rather than adding to it, it bypasses the vocabulary projection and penalty sorting for the noisy branch. On POPE (LLaVA-1.5, RTX 3090), VCD+ACD is 8.2% faster (0.493s vs. 0.537s per example) and uses 3.2% less GPU memory (15.3G vs. 15.8G) compared to standard VCD (detailed in Appendix I).

**Coherence Analysis.** CHAIR requires detailed descriptive generation rather than binary yes/no responses, making it well-suited for evaluating output coherence. Table 7 compares generation quality across Regular, VCD, and VCD+ACD decoding. VCD degrades coherence substantially: CIDEr drops by 13.2% (0.266→0.231), BLEU-4 by 42.6% (0.094→0.054), and GPT-5-assisted scores by 6.5% (0.920→0.860). This stems from logit-level subtraction disrupting vocabulary distributions and penalty terms (Eq. 5) forcing suboptimal token choices. VCD+ACD fully recovers—and slightly exceeds—baseline coherence: CIDEr returns to 0.266, BLEU-4 rises to 0.096 (+2.1% over Regular), and GPT-5 scores reach 0.980 (+6.5%). Two factors drive these gains: (1) attention-level intervention preserves linguistic structure by avoiding abrupt vocabulary shifts; (2) ASS selectively suppresses hallucination-prone regions while amplifying semantic content, yielding more faithful generation. Combined with the hallucination reductions in Table 1 ($Chair_s$: −3.1, $Chair_i$: −4.2), these results confirm that ACD resolves the quality–hallucination trade-off inherent in logit-level contrastive decoding.

*Table 7.* Generation quality evaluation on the CHAIR dataset using LLaVA-1.5 across different decoding strategies.

| Metrics | Regular | VCD | VCD+ACD |
|---|---|---|---|
| **CIDEr↑** | 0.266 | 0.231 | **0.266** |
| **BLEU-4↑** | 0.094 | 0.054 | **0.096** |
| **METEOR↑** | 0.360 | 0.301 | **0.361** |
| **ROUGE-L↑** | 0.365 | 0.324 | **0.368** |
| **GPT-5-assisted↑** | 0.920 | 0.860 | **0.980** |

## 5. Conclusion

This paper tackles the challenge of balancing hallucination suppression with output coherence in LVLMs. We show that existing CD methods degrade generation quality due to logit-level operations and restrictive penalty mechanisms. To address this, we propose ACD, a plug-and-play framework that shifts contrastive operations to the attention layer. The inherent smoothness of attention-level intervention preserves linguistic structure, while ASS dynamically suppresses hallucination-prone attention patterns and reinforces semantic content. Extensive experiments demonstrate that ACD consistently reduces hallucinations across multiple benchmarks without sacrificing—and often improving—coherence metrics. By targeting hallucinations at their attentional source rather than through post-hoc logit adjustment, ACD offers a principled solution that enhances LVLM reliability for real-world deployment.

## Impact Statement

This paper presents work whose goal is to advance the field of Machine Learning. There are many potential societal consequences of our work, none which we feel must be specifically highlighted here.

## Acknowledgements

This work was supported by the Technology Basic Research Project of the State Administration of Science, Technology and Industry for National Defence under Grant JSZL2023416A001.

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

# A. Appendix Overview

This appendix provides a rigorous theoretical foundation, comprehensive empirical validation, and detailed implementation specifics for Attention Contrastive Decoding (ACD). The content is organized as follows:

- **Theoretical Analysis: Smoothness of Attention-Level Operations (Section B):** Establishes the mathematical framework for ACD, providing dimension-independent perturbation bounds and proving why attention-level interventions yield smoother output distributions than logit-level contrastive decoding.

- **Empirical Validation of Distribution Smoothness (Section C):** validates the theoretical propositions through total variation (TV) distance and KL divergence analysis, demonstrating ACD's superior temporal stability and distributional coherence.

- **Robustness to Contrastive Strength (Section D):** Presents ablation studies confirming ACD's stability across varying hyperparameters ($\alpha$), supported by the theoretical scaling analysis.

- **Decoding Procedure (Section E):** Delineates the algorithmic implementation of ACD (Algorithm 1), detailing the dual-attention cache mechanism and the Adaptive Subtraction Strategy.

- **Layer-wise Propagation and Semantic Alignment (Section F):** Analyzes the residual stream dynamics to explain how modality-dependent perturbations propagate through transformer layers while remaining bounded.

- **More Related Work (Section G):** Reviews the landscape of Large Vision-Language Models (LVLMs) and positioning this work within the broader context of hallucination mitigation.

- **Benchmarks and Evaluation Metrics (Section H):** Details the experimental setup, including protocols for CHAIR, LLaVA-Bench, POPE, and MME benchmarks.

- **Efficiency Comparison (Section I):** Benchmarks the computational overhead, demonstrating that ACD reduces latency and memory usage compared to standard VCD.

- **Additional Experimental Results (Sections J & K):** Reports comprehensive performance on the full MME suite and analyzes scalability across 7B and 13B parameter models.

- **Ablation on Sampling Strategies (Section L):** Examines the interaction between ACD and various decoding methods (Greedy, Top-$k$, Top-$p$, etc.), confirming strategy-agnostic improvements.

- **Additional Validation Experiments (Section M):** Presents ASS threshold sensitivity analysis, a stratified analysis of ACD's effect on clean versus hallucinating instances, comprehensive Qwen3-VL (2025) results on MME and POPE, and an analysis of VCD degeneration on deeper architectures.

- **Qualitative Analysis and Setup (Sections N & O):** Provides visual case studies of hallucination correction and details the GPT-5 evaluation prompts used for open-ended assessment.

# B. Theoretical Analysis: Smoothness of Attention-Level Operations

We provide rigorous theoretical justification for why attention-level contrastive operations produce smoother output distributions than logit-level interventions. Our analysis establishes: (1) dimension-independent bounds for attention perturbations versus dimension-dependent bounds for logit perturbations, and (2) the mechanism through which attention perturbations propagate more gracefully to the output distribution. For notational convenience, we denote conventional contrastive decoding at the logit level as CD, and our proposed contrastive decoding at the attention layer as ACD.

## B.1. Perturbation Bounds: Geometric Analysis

We first establish quantitative bounds on perturbation magnitudes in both spaces.

**Definition B.1** (Probability Simplex ([Boyd & Vandenberghe, 2004])). The $(n-1)$-dimensional probability simplex is:

$$\Delta^{n-1} = \left\{ \mathbf{p} \in \mathbb{R}^n : p_i \geq 0, \sum_{i=1}^n p_i = 1 \right\}. \tag{10}$$

**Proposition B.2** (Dimension-Independent Bound for Attention Perturbation). *Let $A_t^{clean}, A_t^{noisy} \in \Delta^{L-1}$ be attention distributions over sequence length $L$. Then:*

$$\|A_t^{clean} - A_t^{noisy}\|_1 \leq 2, \quad \|A_t^{clean} - A_t^{noisy}\|_2 \leq \sqrt{2}. \tag{11}$$

*These bounds are independent of sequence length $L$ and model dimension $d$.*

*Proof.* Partition the index set into $\mathcal{I}^+ = \{i : A_t^{clean}[i] \geq A_t^{noisy}[i]\}$ and $\mathcal{I}^- = \mathcal{I} \setminus \mathcal{I}^+$. Since both distributions sum to unity:

$$\sum_{i \in \mathcal{I}^+} (A_t^{clean}[i] - A_t^{noisy}[i]) = \sum_{i \in \mathcal{I}^-} (A_t^{noisy}[i] - A_t^{clean}[i]). \tag{12}$$

Thus:

$$\|A_t^{clean} - A_t^{noisy}\|_1 = 2 \sum_{i \in \mathcal{I}^+} (A_t^{clean}[i] - A_t^{noisy}[i]) \leq 2 \sum_{i \in \mathcal{I}^+} A_t^{clean}[i] \leq 2. \tag{13}$$

For the $\ell_2$ bound, since $|A_t^{clean}[i] - A_t^{noisy}[i]| \leq 1$ for all $i$:

$$\|A_t^{clean} - A_t^{noisy}\|_2^2 \leq \max_i |A_t^{clean}[i] - A_t^{noisy}[i]| \cdot \|A_t^{clean} - A_t^{noisy}\|_1 \leq 1 \cdot 2 = 2. \tag{14}$$

$\square$

**Proposition B.3** (Dimension-Dependent Bound for Logit Perturbation). *In modern Transformer architectures with final LayerNorm ([Touvron et al., 2023; Brown et al., 2020]), the logit difference satisfies:*

$$\|z_t^{clean} - z_t^{noisy}\|_2 \leq 2\sqrt{d} \cdot \|W_{\text{head}}\|_2, \tag{15}$$

*where $d$ is the hidden dimension and $W_{\text{head}} \in \mathbb{R}^{|\mathcal{V}| \times d}$ is the language model head.*[1]

*Proof.* With LayerNorm applied before the language model head, we have $\|h_t^{clean}\|_2 \approx \sqrt{d}$ and $\|h_t^{noisy}\|_2 \approx \sqrt{d}$. Since $z = W_{\text{head}} h$:

$$\|z_t^{clean} - z_t^{noisy}\|_2 = \|W_{\text{head}}(h_t^{clean} - h_t^{noisy})\|_2 \leq \|W_{\text{head}}\|_2 \cdot \|h_t^{clean} - h_t^{noisy}\|_2. \tag{16}$$

By triangle inequality, $\|h_t^{clean} - h_t^{noisy}\|_2 \leq \|h_t^{clean}\|_2 + \|h_t^{noisy}\|_2 \leq 2\sqrt{d}$. $\square$

*Remark* B.4 (Scaling Comparison). The critical distinction lies in the **scaling behavior**:

- Attention perturbation: $O(1)$, bounded by $\sqrt{2}$ regardless of model scale.
- Logit perturbation: $O(\sqrt{d} \cdot \|W_{\text{head}}\|_2)$, grows with model dimension.

For LLaVA-1.5-7B with $d = 4096$ and typical $\|W_{\text{head}}\|_2 \approx 10$, the logit perturbation bound is $\approx 1280$, approximately three orders of magnitude larger than the attention bound of $\sqrt{2} \approx 1.41$.

---

[1]This assumes standard initialization with normalized variance, under which LayerNorm outputs satisfy $\|h\|_2 \approx \sqrt{d}$ ([Ba et al., 2016]).

## B.2. Propagation Mechanism Analysis

Beyond magnitude bounds, the *mechanism* through which perturbations affect outputs differs fundamentally between CD and ACD.

**Definition B.5** (Contrastive Perturbation Operators). Define the perturbation operators for CD and ACD:

$$\mathcal{P}_{\text{CD}}(z^{clean}, z^{noisy}; \alpha) = (1+\alpha)z^{clean} - \alpha z^{noisy}, \tag{17}$$

$$\mathcal{P}_{\text{ACD}}(A^{clean}, A^{noisy}; \alpha) = (1+\alpha)A^{clean} - \alpha(A^{noisy} \odot M), \tag{18}$$

where $M$ is the adaptive mask from Section 3.3.

**Theorem B.6** (Propagation Path Length). *Let* $\Phi : A^{(l)} \mapsto z$ *denote the forward computation from attention at layer $l$ to final logits. The computational paths differ qualitatively:*

$$\textbf{CD:} \quad z \xrightarrow{\mathcal{P}_{\text{CD}}} z^{CD} \xrightarrow{\text{softmax}} p^{CD}, \tag{19}$$

$$\textbf{ACD:} \quad A^{(l)} \xrightarrow{\mathcal{P}_{\text{ACD}}} A^{ACD} \xrightarrow{\text{Value Agg.}} \xrightarrow{\text{Proj.}} \xrightarrow{\text{LN}} \xrightarrow{\text{FFN}} \xrightarrow{\cdots} \xrightarrow{\text{Head}} z^{ACD} \xrightarrow{\text{softmax}} p^{ACD}. \tag{20}$$

*CD applies perturbation directly before softmax (path length 1), while ACD's perturbation traverses $(N - l + 1)$ layers before affecting output probabilities.*

The extended propagation path in ACD induces two stabilizing effects:

**(1) Subspace Confinement via Value Aggregation.** The attention output is computed as $O = AV$ where $V \in \mathbb{R}^{L \times d_v}$ is the value matrix. Consequently, the output lies in $\text{colspan}(V^\top)$, the column space of $V^\top$. When attention is perturbed by $\Delta A = A^{ACD} - A^{clean}$, the resulting hidden state perturbation satisfies:

$$\Delta h^{(l)} = \Delta A \cdot V \in \text{colspan}(V^\top). \tag{21}$$

**Proposition B.7** (Low-Rank Confinement). *Let* $\text{rank}(V) = r \le \min(L, d_v)$. *Then the attention-induced perturbation* $\Delta h^{(l)}$ *is confined to an $r$-dimensional subspace of $\mathbb{R}^{d_v}$, regardless of the perturbation $\Delta A$.*

*Proof.* Since $\Delta h^{(l)} = \Delta A \cdot V$ and $V$ has rank $r$, we have $\Delta h^{(l)} \in \text{colspan}(V^\top)$, which has dimension exactly $r$. The perturbation cannot escape this subspace regardless of the choice of $\Delta A \in \mathbb{R}^{1 \times L}$. $\qquad\square$

In deep layers where contextualized representations converge, $V$ often exhibits approximate low-rank structure (Wang et al., 2020), further constraining the effective dimensionality of attention perturbations. This geometric confinement has no analogue in logit-level CD, where perturbations can span the full $|\mathcal{V}|$-dimensional vocabulary space.

**(2) LayerNorm Stabilization.** While LayerNorm is not strictly contractive due to the learned scale parameter (Xu et al., 2019), it provides *directional stabilization*: for input $x$ with perturbation $\Delta x$, the first-order approximation yields:

$$\text{LayerNorm}(x + \Delta x) \approx \text{LayerNorm}(x) + \frac{\gamma}{\sigma_x}\Pi_x^\perp(\Delta x), \tag{22}$$

where $\sigma_x$ is the standard deviation of $x$, $\gamma$ is the learned scale, and $\Pi_x^\perp$ projects onto the subspace orthogonal to both $x$ and the all-ones vector. This projection removes two degrees of freedom from $\Delta x$, effectively constraining perturbations to a $(d-2)$-dimensional subspace at each layer (Ba et al., 2016).

## B.3. Softmax Sensitivity and Rank Stability

The most critical distinction concerns how perturbations affect the output probability distribution through the softmax function.

**Lemma B.8** (Softmax Lipschitz Property (Gao & Pavel, 2017)). *For the softmax function $\sigma : \mathbb{R}^n \to \Delta^{n-1}$, the following Lipschitz bound holds:*

$$\|\sigma(z) - \sigma(z')\|_2 \le \|z - z'\|_2. \tag{23}$$

*That is, softmax is 1-Lipschitz with respect to the $\ell_2$ norm.*

**Lemma B.9** (Total Variation via $\ell_2$ Bound). *For probability distributions $p, p' \in \Delta^{n-1}$, the total variation distance satisfies:*

$$\mathrm{TV}(p, p') = \frac{1}{2}\|p - p'\|_1 \leq \frac{\sqrt{n}}{2}\|p - p'\|_2. \tag{24}$$

*Proof.* By definition, $\mathrm{TV}(p, p') = \frac{1}{2}\|p - p'\|_1$. Applying the norm inequality $\|x\|_1 \leq \sqrt{n}\|x\|_2$ for $x \in \mathbb{R}^n$ yields the result. $\qquad\square$

**Theorem B.10** (Rank Inversion Condition in Logit-Level CD). *Consider two tokens $i, j$ with $z_i^{clean} > z_j^{clean}$ (token $i$ preferred under clean input). Under CD with strength $\alpha$, rank inversion ($z_i^{CD} < z_j^{CD}$) occurs if and only if:*

$$z_i^{noisy} - z_j^{noisy} > \frac{1+\alpha}{\alpha}(z_i^{clean} - z_j^{clean}). \tag{25}$$

*Proof.* From $z^{CD} = (1 + \alpha)z^{clean} - \alpha z^{noisy}$:

$$z_i^{CD} < z_j^{CD} \Leftrightarrow (1 + \alpha)(z_i^{clean} - z_j^{clean}) < \alpha(z_i^{noisy} - z_j^{noisy}) \tag{26}$$

$$\Leftrightarrow z_i^{noisy} - z_j^{noisy} > \frac{1+\alpha}{\alpha}(z_i^{clean} - z_j^{clean}). \tag{27}$$

$$\square$$

**Corollary B.11** (Invalid Token Amplification). *For $\alpha = 1$, rank inversion occurs when the noisy preference margin exceeds twice the clean preference margin. As $\alpha \to \infty$, the threshold $\frac{1+\alpha}{\alpha} \to 1$, making rank inversions increasingly likely for any token pair where the noisy and clean preferences disagree.*

*Remark* B.12 (Why ACD Avoids Rank Inversion). ACD perturbations at the attention level do not directly modify logit differences. The mapping $\Phi : A \mapsto z$ is a complex nonlinear composition involving value aggregation (which confines perturbations to a subspace per Proposition B.7), projections, LayerNorms (which provide directional stabilization), and FFNs. Crucially, there exists no simple linear relationship between attention magnitude changes and token logit changes—a token's logit depends on the full sequence context processed through multiple layers, not on any single attention weight. This *decoupling* between attention perturbations and token rankings provides natural robustness against rank inversion.

### B.4. Main Result: Smoothness Guarantee

**Theorem B.13** (Smoothness of ACD Output Distribution). *Let $p^{vanilla}, p^{CD}, p^{ACD} \in \Delta^{|\mathcal{V}|-1}$ denote the output token distributions under vanilla decoding, logit-level CD, and attention-level ACD, respectively. Define the distribution shifts:*

$$\Delta_{\mathrm{CD}} = \mathrm{TV}(p^{CD}, p^{vanilla}), \quad \Delta_{\mathrm{ACD}} = \mathrm{TV}(p^{ACD}, p^{vanilla}). \tag{28}$$

*Then:*

*(I) [Direct Bound for CD] Combining Lemmas B.8 and B.9:*

$$\begin{aligned}
\Delta_{\mathrm{CD}} &= \frac{1}{2}\|p^{CD} - p^{vanilla}\|_1 \leq \frac{\sqrt{|\mathcal{V}|}}{2}\|p^{CD} - p^{vanilla}\|_2 \\
&\leq \frac{\sqrt{|\mathcal{V}|}}{2}\|z^{CD} - z^{clean}\|_2 = \frac{\alpha\sqrt{|\mathcal{V}|}}{2}\|z^{clean} - z^{noisy}\|_2 \\
&\leq \alpha\sqrt{|\mathcal{V}|} \cdot \sqrt{d} \cdot \|W_{\mathrm{head}}\|_2 = O(\alpha\sqrt{d \cdot |\mathcal{V}|}),
\end{aligned} \tag{29}$$

*where the final inequality applies Proposition B.3.*

*(II) [Structural Bound for ACD] The ACD perturbation propagates through $\Phi$:*

$$\Delta_{\mathrm{ACD}} \leq \frac{\sqrt{|\mathcal{V}|}}{2}\|z^{ACD} - z^{clean}\|_2 = \frac{\sqrt{|\mathcal{V}|}}{2}\|\Phi(A^{ACD}) - \Phi(A^{clean})\|_2. \tag{30}$$

*While $\Phi$ involves residual connections that preclude global contraction guarantees, the input perturbation satisfies the dimension-independent bound $\|A^{ACD} - A^{clean}\|_2 \leq \alpha\sqrt{2}$ (Proposition B.2), and propagation through value aggregation confines perturbations to a low-rank subspace (Proposition B.7). Combined with LayerNorm stabilization, these mechanisms yield empirically observed smoothness $\Delta_{\mathrm{ACD}} \ll \Delta_{\mathrm{CD}}$, validated in Section C.*

**Corollary B.14** (Elimination of Penalty Mechanisms). *The bounded attention perturbation (Proposition B.2) combined with the subspace confinement (Proposition B.7) and indirect propagation mechanism (Theorem B.6) ensures that ACD does not cause severe rank inversions (Theorem B.10). Consequently, the penalty mechanism in Equation 5—designed to filter invalid tokens arising from rank inversions—becomes unnecessary under ACD, enabling simpler deployment without hyperparameter tuning for $\beta$.*

**Corollary B.15** (Robustness to Contrastive Strength). *Since $\|A^{ACD} - A^{clean}\|_2 \leq \alpha\sqrt{2}$ scales linearly with $\alpha$ with a small, dimension-independent coefficient ($\sqrt{2} \approx 1.41$), ACD maintains stable performance across varying $\alpha$ values. In contrast, CD's perturbation $\alpha\|z^{clean} - z^{noisy}\|_2$ has a dimension-dependent coefficient that can reach $O(\sqrt{d})$, amplifying instability at high $\alpha$. This analysis is consistent with empirical observations in Figure 6.*

*Remark* B.16 (Information-Geometric Perspective). The probability simplex $\Delta^{n-1}$ equipped with the Fisher-Rao metric exhibits curvature that increases near boundaries where some $p_i \to 0$. This intrinsic geometry implicitly penalizes perturbations that push attention toward extreme values, providing geometric regularization absent in the flat Euclidean structure of logit space.

# C. Empirical Validation of Distribution Smoothness and Decoding Coherence

Our theoretical analysis in Section B establishes that attention-level contrastive operations yield smoother output distributions than logit-level interventions, owing to dimension-independent perturbation bounds (Proposition B.2), low-rank subspace confinement (Proposition B.7), and the extended propagation path through stabilizing architectural components (Theorem B.6). In this section, we provide comprehensive empirical evidence corroborating these theoretical predictions.

## C.1. Experimental Protocol

We evaluate the distributional smoothness of VCD (logit-level contrastive decoding) and VCD+ACD (attention-level contrastive decoding applied on top of VCD) using LLaVA-1.5-7B (Liu et al., 2023b) on the CHAIR benchmark (Yue et al., 2024). We sample 500 images and generate captions with a maximum token length of 128. At each decoding step $t$, we record the output probability distribution and compute two complementary divergence measures relative to the vanilla (unperturbed) decoding distribution $p_t^{\text{vanilla}}$:

- **Total Variation (TV) Distance**: $\text{TV}(p_t, p_t^{\text{vanilla}}) = \frac{1}{2}\|p_t - p_t^{\text{vanilla}}\|_1$, which directly measures the maximum difference in event probabilities and corresponds to the theoretical bounds in Theorem B.13.

- **KL Divergence**: $D_{\text{KL}}(p_t^{\text{vanilla}}\|p_t) = \sum_i p_t^{\text{vanilla}}(i) \log \frac{p_t^{\text{vanilla}}(i)}{p_t(i)}$, which captures information-theoretic distortion and is particularly sensitive to probability mass assigned to unlikely tokens.

We visualize the results through four complementary perspectives: (1) aggregate box plots summarizing overall distributional statistics, (2) violin plots revealing the full density shape of divergence values, and (3,4) step-wise trajectory plots with $\pm 1$ standard deviation bands illustrating temporal dynamics across the decoding process.

## C.2. Results and Analysis

**Overall Distribution Shift (TV Distance).** As shown in Figures 4a and 4b, VCD+ACD achieves a dramatic reduction of **79.9%** in mean TV distance compared to VCD ($\mu = 0.131$ vs. $\mu = 0.653$). The box plot reveals that VCD's interquartile range (IQR) spans approximately $[0.49, 0.87]$ with the median at $0.72$, indicating that logit-level subtraction pervasively distorts the output distribution—over half of all decoding steps exhibit a TV distance exceeding $0.7$, signifying that the top-ranked token under VCD frequently differs from that of vanilla decoding. In stark contrast, VCD+ACD's IQR is compressed to approximately $[0.04, 0.19]$ with a median of $0.11$, confirming that attention-level perturbations preserve the overall distributional structure with minimal deviation.

The violin plot further elucidates the density profile: VCD's distribution mass concentrates in the high-divergence regime $(0.6$–$1.0)$ with a pronounced mode near $1.0$—the theoretical maximum of TV distance—indicating frequent near-complete redistribution of probability mass. Conversely, VCD+ACD exhibits a sharply peaked distribution concentrated near zero, with only a thin tail extending beyond $0.4$. This empirical observation directly validates Theorem B.13: the dimension-independent bound on attention perturbations ($\|A^{ACD} - A^{clean}\|_2 \le \alpha\sqrt{2}$) combined with low-rank confinement and LayerNorm stabilization yields substantially smaller distributional shifts compared to the dimension-dependent logit perturbation bound of $O(\alpha\sqrt{d \cdot |\mathcal{V}|})$.

**Information-Theoretic Distortion (KL Divergence).** Figures 4c and 4d present the KL divergence analysis. VCD+ACD reduces mean KL divergence by **10.7%** ($\mu = 1.927$ vs. $\mu = 2.157$). While this relative reduction appears modest compared to the TV distance improvement, two critical observations merit emphasis. First, VCD exhibits substantially heavier tails: the box plot shows numerous outliers exceeding 10 nats, with extreme values reaching 21.1 nats, whereas VCD+ACD's outliers are capped at approximately 11.2 nats. The violin plot corroborates this finding—VCD's density exhibits a visible secondary mode in the 12–18 range, corresponding to decoding steps where logit-level subtraction catastrophically distorts the probability distribution. Such extreme KL values indicate that VCD occasionally assigns near-zero probability to tokens favored by vanilla decoding, a pathological behavior consistent with the rank inversion phenomenon characterized in Theorem B.10. Second, VCD+ACD's IQR is more compact (approximately $[0.5, 3.0]$ vs. $[0.8, 3.5]$), reflecting more predictable and controlled distributional modifications.

The asymmetry between the TV and KL improvements is theoretically expected: KL divergence is dominated by tail behavior ($\log \frac{p}{q}$ diverges as $q \to 0$), and both methods occasionally produce small probability perturbations that inflate KL

values. However, TV distance, which corresponds to the $\ell_1$ norm of the probability difference, is more faithful to the overall distributional shape—precisely the quantity bounded by our theoretical analysis.

**Temporal Dynamics Across Decoding Steps.** The step-wise trajectory plots (Figures 4e and 4f) reveal the temporal coherence of both methods throughout the autoregressive generation process.

For TV distance (Figure 4e), VCD exhibits consistently high distortion (0.5–0.7) with a large standard deviation band spanning nearly the entire $[0, 1]$ range, indicating highly variable and unpredictable distributional shifts across both time steps and samples. Notably, VCD's mean TV distance increases slightly over time (from $\sim$0.5 at step 10 to $\sim$0.7 at step 128), suggesting that distributional perturbations accumulate as the generated sequence grows longer—a consequence of error propagation in autoregressive decoding where each distorted token conditions subsequent generation. In contrast, VCD+ACD maintains remarkably stable TV distances (0.1–0.2) with a narrow standard deviation band throughout the entire decoding horizon. This temporal stability empirically validates Corollary B.15: since attention perturbations are bounded by a dimension-independent constant at every layer and every step, the resulting output distribution remains consistently close to the vanilla distribution regardless of sequence length.

For KL divergence (Figure 4f), both methods exhibit an initial transient phase (steps 1–5) with elevated divergence, attributable to the high uncertainty in early-stage token prediction where the context window is minimal. After this transient, VCD+ACD converges to a lower and more stable trajectory ($\sim$1.5–2.0 nats) compared to VCD ($\sim$2.0–2.5 nats). Crucially, VCD's standard deviation band remains substantially wider throughout, confirming that logit-level subtraction introduces heterogeneous perturbation magnitudes that vary unpredictably across decoding steps—precisely the instability predicted by the dimension-dependent bound in Proposition B.3.

**Summary.** The empirical results across all six diagnostic visualizations converge on a consistent conclusion: *ACD produces fundamentally smoother, more coherent, and more temporally stable output distributions than logit-level contrastive decoding*. The 79.9% reduction in TV distance validates our theoretical prediction that attention-level operations, constrained by dimension-independent bounds and low-rank subspace confinement, preserve the model's intrinsic generation distribution far more faithfully than logit-level interventions. The temporal trajectory analysis further demonstrates that this smoothness advantage is not merely an aggregate statistical artifact but persists consistently across every decoding step, ensuring coherent and stable generation throughout the entire output sequence. These findings provide strong empirical support for Theorem B.13 and Corollary B.14, substantiating that ACD's architectural design eliminates the need for auxiliary penalty mechanisms while maintaining superior distributional fidelity.

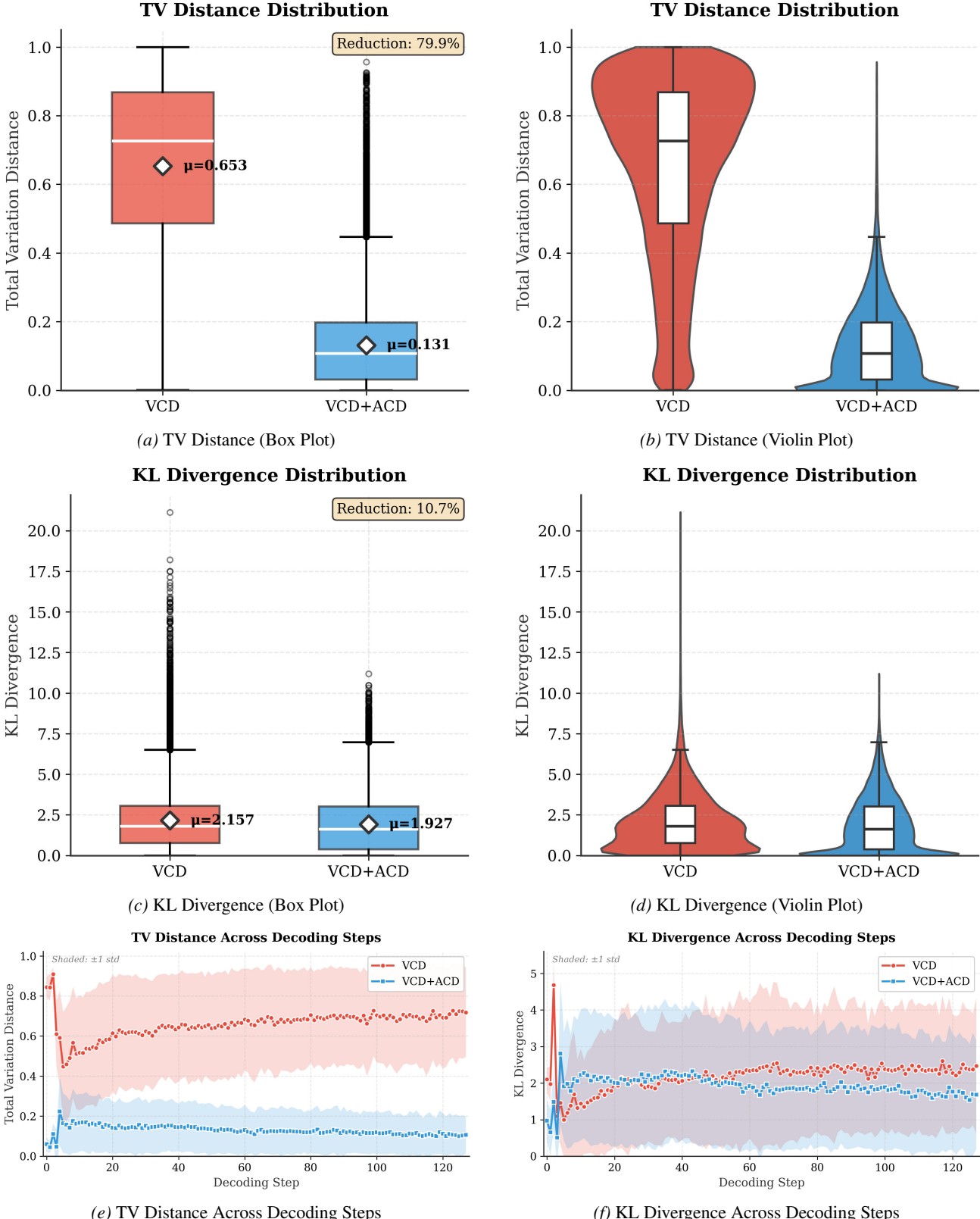

*Figure 4.* **Empirical validation of distributional smoothness.** We compare VCD (logit-level) and VCD+ACD (attention-level) on LLaVA-1.5-7B across 500 CHAIR images (max 128 tokens). (a,b) TV distance distributions; (c,d) KL divergence distributions; (e,f) step-wise temporal dynamics with ±1 std shading. ACD consistently yields smoother, more stable output distributions across all metrics and decoding steps.

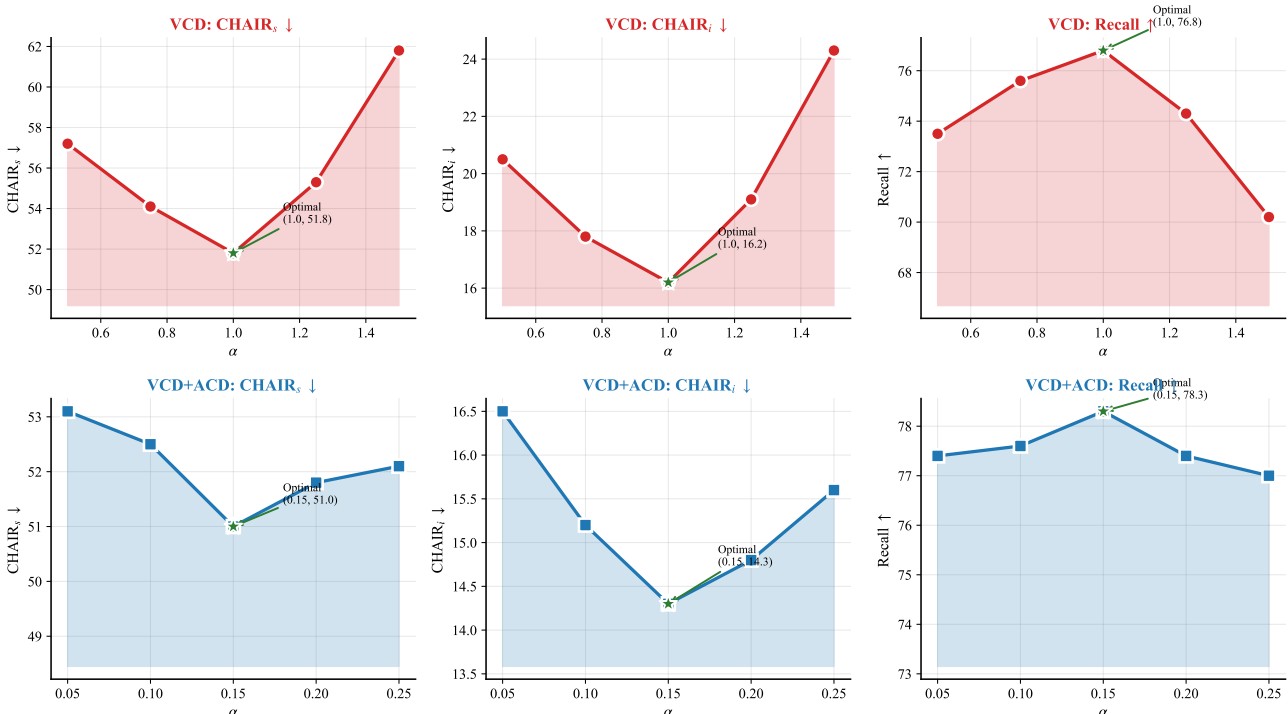

*Figure 5.* **Absolute performance across $\alpha$ values.** Top: VCD ($\alpha \in [0.5, 1.5]$); Bottom: VCD+ACD ($\alpha \in [0.05, 0.25]$). Green stars indicate optimal configurations. VCD exhibits substantial variation while VCD+ACD maintains tight performance bounds.

## D. Robustness to Contrastive Strength $\alpha$

Our theoretical analysis (Corollary B.15) predicts that ACD exhibits superior robustness to the contrastive strength hyperparameter $\alpha$ due to its dimension-independent perturbation bound. We empirically validate this prediction by conducting ablation studies on both VCD and VCD+ACD using LLaVA-1.5-7B on the CHAIR benchmark.

**Results.** Figure 5 presents performance trajectories across $\alpha$ values. VCD achieves optimal performance at $\alpha = 1.0$ but degrades substantially at non-optimal values—at $\alpha = 1.5$, CHAIR$_i$ increases by 50% (from 16.2 to 24.3). In contrast, VCD+ACD maintains remarkably stable performance across its entire evaluated range ($\alpha \in [0.05, 0.25]$), with CHAIR$_i$ varying by only 2.2 points.

Figure 6 normalizes $\alpha$ to each method's optimum for direct comparison. VCD exhibits steep degradation curves, with performance declining rapidly as $\alpha$ deviates from optimal. VCD+ACD displays a nearly flat response, with degradation remaining below 5% even at boundary values. Quantitatively, VCD+ACD achieves approximately **78%** lower standard deviation across all metrics compared to VCD.

This contrast directly reflects the theoretical distinction: VCD's perturbation scales as $O(\alpha\sqrt{d})$ with a large dimension-dependent coefficient, while ACD's perturbation scales as $O(\alpha)$ with coefficient $\sqrt{2} \approx 1.41$. The dimension-independent bound acts as a natural regularizer, ensuring stable performance regardless of $\alpha$ selection.

## E. Decoding Procedure

The specific algorithm is as follows in Algorithm 1.

During practical decoding, the ACD method maintains dual attention caches: an original cache and a hallucination cache. At each decoding step, ACD computes attention distributions for both original and hallucination-inducing inputs, subsequently applying the ASS strategy to identify and suppress attention allocations that amplify important semantic information potentially conducive to hallucination generation. This fine-grained intervention at the attention layer enables the model to maintain linguistic coherence and diversity while effectively mitigating hallucinations. In contrast to traditional CD

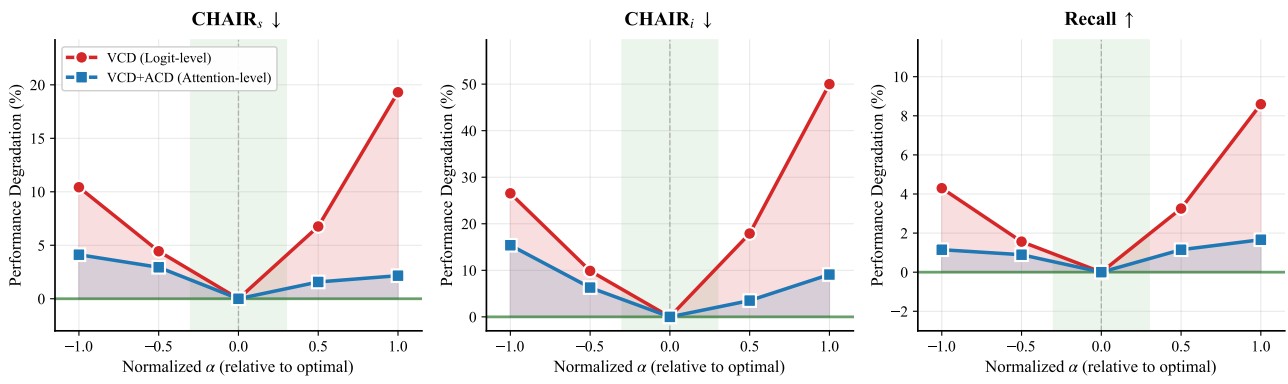

*Figure 6.* **Normalized robustness comparison.** Performance degradation (%) versus normalized $\alpha$ (centered at optimal). VCD (red) exhibits steep degradation; VCD+ACD (blue) maintains near-flat response, validating Corollary B.15.

---

**Algorithm 1** Attention Contrastive Decoding (ACD)

---

1: **Input:** Image $v$, text query $x$, model $\theta$, hyperparameters $\alpha$
2: **Choose:** Perturbation function $\mathcal{P}(\cdot)$ to generate $(v', x')$
3: **Output:** Generated sequence $y = [y_1, y_2, \ldots, y_T]$
4: Initialize: $y_{<1} \leftarrow \emptyset$, KV caches: $\mathcal{K}^{clean}, \mathcal{K}^{noisy} \leftarrow \emptyset$
5: **for** $t = 1, 2, \ldots, T$ **do**
6:     **// Parallel Forward Pass**
7:     Compute query: $q_t \leftarrow \text{Query}(v, x, y_{<t})$
8:     **// Clean path (original inputs)**
9:     $A_t^{clean} \leftarrow \text{Softmax}\left(\frac{q_t K_{\leq t}^{clean\top}}{\sqrt{d}}\right)$                        *Eq. (7)*
10:     **// Noisy path (hallucination-inducing inputs)**
11:     $(v', x') \leftarrow \mathcal{P}(v, x)$                         *e.g., VCD, ICD, SID*
12:     $A_t^{noisy} \leftarrow \text{Softmax}\left(\frac{q_t K_{\leq t}^{noisy\top}}{\sqrt{d}}\right)$                        *Eq. (7)*
13:     **// Adaptive Subtraction Strategy (ASS)**
14:     $M_t \leftarrow \mathbb{I}(A_t^{noisy} > A_t^{clean})$
15:     $A_t^{ACD} \leftarrow (1 + \alpha)A_t^{clean} - \alpha(A_t^{noisy} \odot M_t)$            *Eq. (9)*
16:     **// Logit Computation with Modified Attention**
17:     $\text{logit}_\theta(y_t|v, x, y_{<t}) \leftarrow \text{LLM}(A_t^{ACD} V_{\leq t}^{clean})$
18:     **// Sampling and Updating**
19:     $y_t \sim \text{Softmax}(\text{logit}_\theta(y_t|v, x, y_{<t}))$
20:     Update KV caches: $K_{\leq t+1}^{clean}$ and $K_{\leq t+1}^{noisy}$
21: **end for**
22: **Return** $y = [y_1, y_2, \ldots, y_T]$

---

methods, ACD eliminates the necessity for $\mathcal{V}_{\text{head}}$ penalty mechanisms as formulated in Equation 5 to filter low-probability tokens, as its attention-layer intervention inherently prevents invalid token generation. This characteristic allows ACD to reduce hallucinations while preserving content quality and diversity. The fundamental advantage of ACD lies in its exploitation of the inherent smoothness and structural continuity of the Transformer attention mechanism, enabling the model to precisely localize and suppress hallucination sources while maintaining generation quality. This approach provides a more reliable decoding solution for practical LVLM applications, effectively balancing hallucination mitigation with output quality preservation.

# F. Layer-wise Propagation and Semantic Alignment

We analyze how the perturbation introduced in the noisy stream propagates through the transformer layers and establish that the resulting divergence remains bounded—providing the theoretical foundation for ACD's contrastive mechanism.

## F.1. Residual Stream Dynamics

Consider a transformer with $L$ layers. Let $\mathbf{h}_l^{(c)}$ and $\mathbf{h}_l^{(n)}$ denote the hidden states at layer $l$ for the clean and noisy streams, respectively. The residual update rule gives:

$$\mathbf{h}_{l+1}^{(c)} = \mathbf{h}_l^{(c)} + f_l(\mathbf{h}_l^{(c)}), \quad \mathbf{h}_{l+1}^{(n)} = \mathbf{h}_l^{(n)} + f_l(\mathbf{h}_l^{(n)}), \tag{31}$$

where $f_l(\cdot)$ denotes the combined self-attention and feed-forward transformation at layer $l$. Define the inter-stream divergence as $\boldsymbol{\delta}_l = \mathbf{h}_l^{(c)} - \mathbf{h}_l^{(n)}$. Its recurrence is:

$$\boldsymbol{\delta}_{l+1} = \boldsymbol{\delta}_l + \left[ f_l(\mathbf{h}_l^{(c)}) - f_l(\mathbf{h}_l^{(n)}) \right]. \tag{32}$$

**Proposition F.1** (Bounded Cumulative Divergence). *Assume each layer transformation $f_l$ is Lipschitz continuous with constant $K_l < 1$. Then the divergence satisfies:*

$$\|\boldsymbol{\delta}_{l+1}\| \leq \|\boldsymbol{\delta}_l\| + K_l\|\boldsymbol{\delta}_l\| = (1 + K_l)\|\boldsymbol{\delta}_l\|. \tag{33}$$

*Unrolling through all layers yields:*

$$\|\boldsymbol{\delta}_L\| \leq \|\boldsymbol{\delta}_0\| \prod_{l=0}^{L-1} (1 + K_l). \tag{34}$$

*When $K_l \ll 1$ (as enforced by LayerNorm and residual scaling), $\prod_l(1 + K_l) \approx \exp\left(\sum_l K_l\right)$, so the divergence grows at most* exponentially in the sum of Lipschitz constants *rather than unboundedly.*

In practice, the pre-trained LLM satisfies a stronger condition. Denoting the cosine similarity $s_l = \cos(\mathbf{h}_l^{(c)}, \mathbf{h}_l^{(n)})$, we can decompose the per-layer similarity change as:

$$s_{l+1} = \frac{\langle \mathbf{h}_l^{(c)} + f_l(\mathbf{h}_l^{(c)}), \ \mathbf{h}_l^{(n)} + f_l(\mathbf{h}_l^{(n)}) \rangle}{\|\mathbf{h}_l^{(c)} + f_l(\mathbf{h}_l^{(c)})\| \ \cdot \ \|\mathbf{h}_l^{(n)} + f_l(\mathbf{h}_l^{(n)})\|}. \tag{35}$$

Expanding the numerator:

$$\langle \mathbf{h}_{l+1}^{(c)}, \mathbf{h}_{l+1}^{(n)} \rangle = \langle \mathbf{h}_l^{(c)}, \mathbf{h}_l^{(n)} \rangle + \langle \mathbf{h}_l^{(c)}, f_l(\mathbf{h}_l^{(n)}) \rangle$$
$$+ \langle f_l(\mathbf{h}_l^{(c)}), \mathbf{h}_l^{(n)} \rangle + \langle f_l(\mathbf{h}_l^{(c)}), f_l(\mathbf{h}_l^{(n)}) \rangle. \tag{36}$$

The residual term $\langle \mathbf{h}_l^{(c)}, \mathbf{h}_l^{(n)} \rangle$ dominates the cross terms when $\|f_l\| \ll \|\mathbf{h}_l\|$, which prevents $s_l$ from collapsing to zero even across many layers. This explains the *bounded, non-monotonic* similarity profile observed empirically.

**Corollary F.2** (Modality-Dependent Divergence). *Let $\mathcal{T}$ and $\mathcal{V}$ denote the index sets of text and visual tokens. Since the two streams share identical text inputs but differ in visual inputs, the initial divergence satisfies $\|\boldsymbol{\delta}_0^{(\mathcal{T})}\| = 0$ and $\|\boldsymbol{\delta}_0^{(\mathcal{V})}\| > 0$. By Proposition F.1, for all layers $l$:*

$$\mathbb{E}\big[s_l^{(\mathcal{T})}\big] \ > \ \mathbb{E}\big[s_l^{(\mathcal{V})}\big], \tag{37}$$

*i.e., text tokens consistently maintain higher inter-stream similarity than visual tokens. The gap $\Delta_l = \mathbb{E}[s_l^{(\mathcal{T})}] - \mathbb{E}[s_l^{(\mathcal{V})}]$ quantifies the modality-dependent perturbation signal that ACD exploits.*

## F.2. Empirical Validation

We measure per-layer cosine similarity between clean and noisy hidden states on 100 randomly sampled CHAIR instances (noise step $T$=800). Figure 7 reports the mean and standard deviation across samples.

Three findings emerge from the empirical results:

**(1) Bounded, non-monotonic divergence.** The overall similarity does not decay monotonically. It reaches a minimum of 0.59 around layer 18 before recovering in deeper layers, confirming Proposition F.1—residual connections prevent

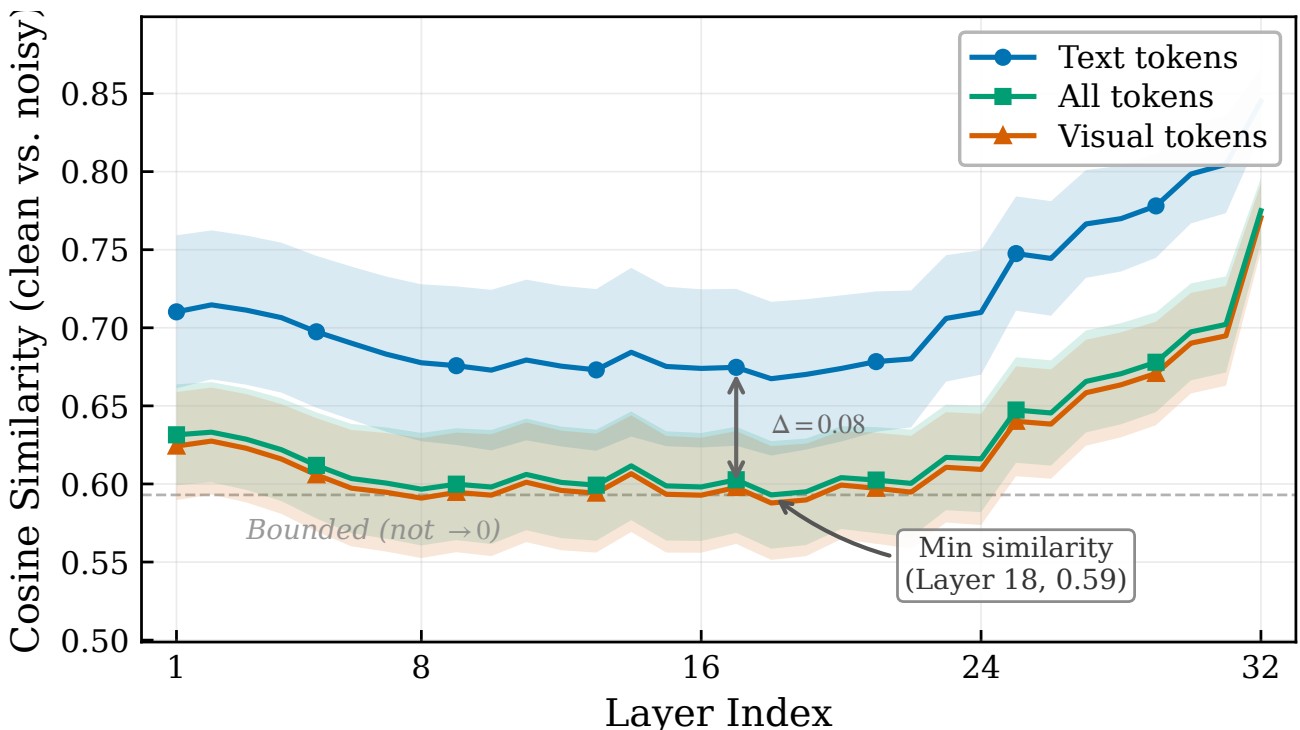

*Figure 7.* Layer-wise cosine similarity between clean and noisy stream hidden states (LLaVA-1.5-7B, 32 layers). Text tokens maintain consistently higher similarity than visual tokens. Divergence peaks at middle layers and partially recovers in deeper layers, confirming bounded propagation.

unbounded divergence accumulation. Across all 32 layers, similarity remains above $0.59$, indicating that the two streams never become semantically orthogonal.

**(2) Modality gap.** Text tokens maintain a mean similarity of $0.71 \pm 0.04$, while visual tokens show $0.62 \pm 0.04$, yielding a consistent gap $\Delta \approx 0.09$ (Corollary F.2). This gap arises because text tokens receive identical embeddings in both streams, whereas visual token representations are directly perturbed. The gap persists across all layers, providing a stable contrastive signal.

**(3) Informative divergence regime.** The moderate similarity range ($0.59$–$0.71$) is neither too high nor too low. If streams were near-identical ($s_l > 0.95$), ACD would have no contrastive signal to exploit; if they were orthogonal ($s_l \approx 0$), the noisy stream would carry no semantically meaningful structure. The observed regime provides sufficient contrast to identify hallucination-prone attention patterns while preserving the shared semantic backbone for valid reasoning. This validates our design choice of noise step $T{=}800$, which places the perturbation in this *informative divergence regime*.

## G. More RELATED WORK

### G.1. Large Vision-Language Models

Large Vision-Language Models (LVLMs)(Liu et al., 2023b;c; 2024a; Zhu et al., 2023; Bai et al., 2023; Li et al., 2023a; Driess et al., 2023; Achiam et al., 2023; Chowdhery et al., 2023; Alayrac et al., 2022) have emerged as a pivotal research direction, effectively bridging computer vision and natural language processing paradigms. These models demonstrate exceptional capabilities across diverse multimodal tasks, including image captioning(Lin et al., 2014; Li et al., 2023a), visual question answering (VQA)(Liu et al., 2023c;b; 2024a; Hudson & Manning, 2019), and sophisticated multimodal reasoning(Lu et al., 2022; Alayrac et al., 2022; Achiam et al., 2023). Modern LVLM architectures typically integrate visual encoders (Tong et al., 2024) with feature projection modules (Liu et al., 2023c; 2024a; Zhu et al., 2023) that interface with large language models (LLMs) (Radford et al., 2018; Devlin et al., 2019; Brown et al., 2020; Raffel et al., 2020; Chowdhery et al., 2023; Thoppilan et al., 2022; Achiam et al., 2023), creating a unified embedding space where visual and textual representations converge to enable sophisticated cross-modal understanding. Despite significant advances in LVLMs, hallucination phenomena (Li

et al., 2023c; Fu et al., 2023b; Yue et al., 2024)—wherein models generate content fundamentally inconsistent with visual inputs—remain a critical limitation. This work addresses the challenge of hallucination mitigation in contemporary LVLMs, aiming to enhance their reliability and expand their applicability across diverse domains.

## G.2. Broader Context

Adaptive feature recalibration and robust representation learning are fundamental challenges in visual perception. A growing body of work addresses representational fidelity under distributional shift (Li et al., 2026; Chen et al., 2026a;b; Tao et al., 2026; Chen et al., 2024a; Deng et al., 2025; Chen et al., 2025d), developing adaptive strategies to preserve semantically meaningful signals amid noisy or out-of-distribution inputs. Balancing competing objectives without sacrificing output quality has also attracted considerable attention (Sun et al., 2025b;a). In the context of cross-modal understanding, selective and structure-preserving alignment between heterogeneous representations remains an active research direction (Cheng et al., 2025b;a;c; 2026d;c;b;a; Cheng, 2026; Cheng et al., 2024), as does training-free adaptive inference for vision-language models (Chen et al., 2025b).

# H. Benchmarks and Evaluation Metric Details

This section delineates the comprehensive evaluation framework employed in our experimental analysis, encompassing four distinct benchmark datasets and their corresponding evaluation protocols designed to systematically assess hallucination mitigation efficacy across varied visual reasoning paradigms.

## H.1. CHAIR

CHAIR (Yue et al., 2024) (Caption Hallucination Assessment with Image Relevance) quantifies object hallucination severity in generated image descriptions. Unlike conventional metrics that evaluate image descriptions through lexical or syntactic similarity, CHAIR directly leverages image annotation metadata to identify hallucination instances, thereby measuring semantic consistency between generated descriptions and actual visual content with greater precision. The CHAIR framework comprises two complementary metrics:

- $CHAIR_s$ (sentence-level): Measures the proportion of generated sentences containing at least one hallucinated object

- $CHAIR_i$ (instance-level): Quantifies the ratio of hallucinated object mentions to total object mentions across all generated descriptions

Additionally, recall serves as a complementary metric to evaluate semantic comprehensiveness, measuring the proportion of ground-truth objects successfully captured in the generated descriptions. This methodological approach circumvents potential biases inherent in purely linguistic matching techniques, offering enhanced sensitivity in hallucination detection while revealing semantic fidelity deficiencies that might remain undetected through traditional evaluation frameworks.

$$CHAIR_s = \frac{|\{\text{Captions with hallucinated objects}\}|}{|\{\text{All captions}\}|} \tag{38}$$

$$CHAIR_i = \frac{|\{\text{Hallucinated objects}\}|}{|\{\text{All mentioned objects}\}|} \tag{39}$$

$$Recall = \frac{|\{\text{Accurate objects}\}|}{|\{\text{Ground-truth objects}\}|} \tag{40}$$

## H.2. LLaVA-Bench (In-the-Wild)

LLaVA-Bench (In-the-Wild) (Liu et al., 2023b) encompasses diverse challenging visual scenarios including indoor/outdoor environments, internet memes, paintings, sketches, and various artistic or abstract imagery typically underrepresented in standard training distributions. This benchmark deliberately introduces distributional shifts to assess model generalization capabilities across challenging visual contexts. The dataset comprises 24 distinct images with 60 corresponding questions, each image accompanied by meticulously crafted comprehensive descriptions and multiple targeted questions. Following an

instruction-following evaluation paradigm, models receive visual inputs paired with natural language instructions requiring appropriate responses. The assessment protocol incorporates questions spanning fundamental visual comprehension to complex inferential reasoning. Given the absence of standardized ground truth evaluations, we employ GPT-5 (OpenAI, 2025), currently the most advanced Large Vision-Language Model available, as an automatic evaluator for generated descriptions. Through carefully engineered prompting (illustrated in Figure 13), the evaluation assesses model outputs along two critical dimensions: (1) Accuracy—measuring semantic consistency between generated descriptions and visual content; and (2) Detailedness—quantifying the richness and comprehensiveness of visual details captured in model-generated descriptions. This automated evaluation methodology enables consistent assessment of model performance across challenging out-of-distribution visual scenarios.

### H.3. POPE

POPE (Li et al., 2023c) (Polling-based Object Presence Evaluation) formalizes the assessment of object hallucination as a binary classification task by prompting LVLMs with yes/no questions regarding object presence in images. Each evaluation sample consists of a triplet containing an image, corresponding question, and ground-truth answer. The framework constructs both positive and negative question instances with varying difficulty levels through three distinct sampling strategies: Random sampling—selecting arbitrarily from objects absent in the image; Popular sampling—selecting the top $\frac{l}{2}$ most frequent objects across the dataset that are absent from the current image; and Adversarial sampling—selecting the top $k$ absent objects ranked by co-occurrence frequency with objects present in the image. Our POPE evaluation dataset incorporates over 20,000 question-answer pairs constructed from MSCOCO (Lin et al., 2014), A-OKVQA (Schwenk et al., 2022), and GQA (Hudson & Manning, 2019) datasets. Within the POPE evaluation framework, model responses are constrained to binary "Yes" or "No" outputs, effectively transforming object hallucination detection into a standard binary classification problem. Consequently, we employ established classification metrics—Accuracy, Precision, Recall, and F1-score—to quantitatively assess model performance. These metrics are formally defined as:

$$\text{Accuracy} = \frac{TP + TN}{TP + TN + FP + FN}, \tag{41}$$

$$\text{Precision} = \frac{TP}{TP + FP}, \tag{42}$$

$$\text{Recall} = \frac{TP}{TP + FN}, \tag{43}$$

$$\text{F1-score} = \frac{2 \cdot \text{Precision} \cdot \text{Recall}}{\text{Precision} + \text{Recall}}, \tag{44}$$

where $TP$, $TN$, $FP$, and $FN$ denote true positive, true negative, false positive, and false negative classifications, respectively. $Accuracy$ quantifies the proportion of correct predictions across all instances; $Precision$ measures the model's ability to avoid false positives when identifying present objects; $Recall$ assesses the model's capacity to identify all actually present objects; and $F1 - score$ provides a harmonic mean of Precision and Recall, offering a comprehensive performance metric that balances both dimensions of classification quality.

### H.4. MME

MME (Fu et al., 2023a) (Multimodal Model Evaluation) constitutes a comprehensive benchmark for multimodal large language model assessment, encompassing 14 distinct subtasks categorized into perceptual domains (object existence, quantification, color identification, spatial relationships) and cognitive domains (commonsense reasoning, numerical computation, text translation). To facilitate quantitative assessment, MME employs a standardized binary response format analogous to POPE, requiring "Yes/No" responses. A distinctive characteristic of this framework is its complementary instruction design—each visual input is paired with dual instructions, one requiring an affirmative response and the other a negative response based on ground truth. For evaluation metrics, MME implements an $accuracy$-based assessment methodology similar to POPE, calculating performance based on correct classification across individual instructions.

*Table 8.* A comparison of inference speed and GPU memory usage for different methods applied to the LLaVA-1.5 model on POPE subset benchmark.

| Method | Total Time | GPU-Memory | Latency/Example |
|---|---|---|---|
| Regular | 13:46 | 14.7G | 0.276s |
| VCD | 26:51 | 15.8G | 0.537s |
| **VCD+ACD** | **24:37** | **15.3G** | **0.493s** |

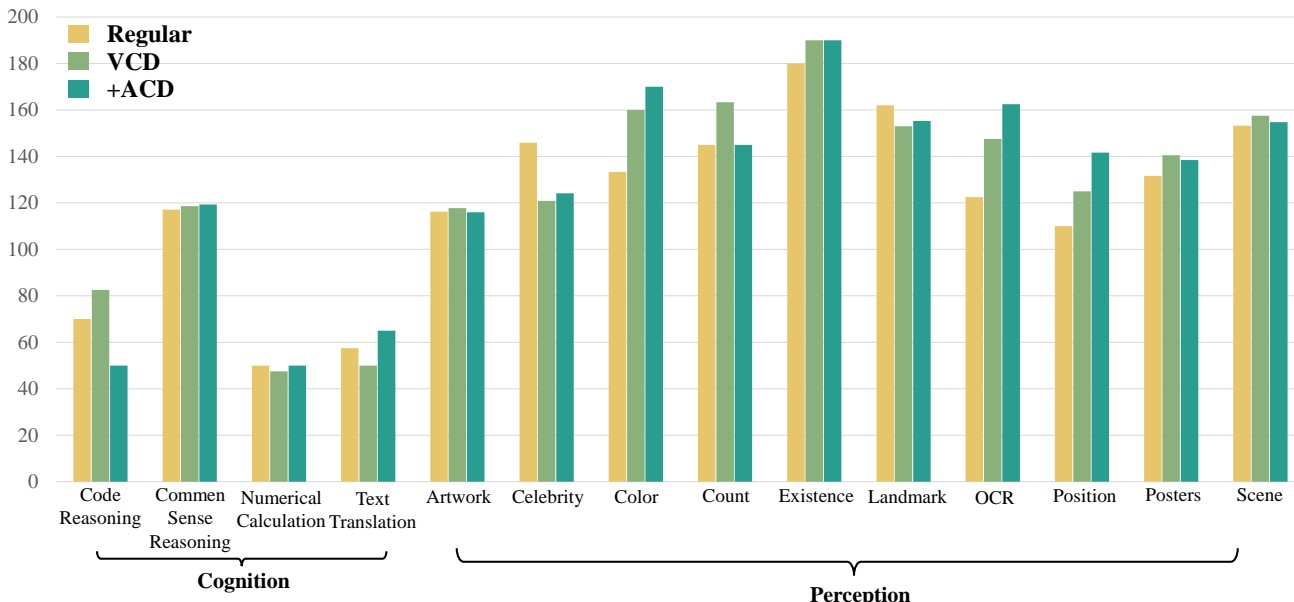

*Figure 8.* MME full set results on LLaVA-1.5

## I. Efficiency Comparison

We benchmark inference efficiency on the POPE MSCOCO Random split using LLaVA-1.5 on a single NVIDIA RTX 3090 24GB GPU. As shown in Table 8, CD methods roughly double the runtime of regular decoding while incurring only modest additional memory usage. Compared with VCD, VCD+ACD reduces both latency (0.537s→0.493s per example) and GPU memory (15.8G→15.3G). The savings arise because ACD operates solely on attention maps for hallucination-prone inputs, bypassing the full forward pass through hidden states that VCD requires. This targeted computation enables stronger hallucination suppression at lower overhead, making ACD a more efficient plug-in for practical LVLM deployment.

## J. MME Full Set Results

Figure 8 reports scores on the complete MME benchmark using LLaVA-1.5, which exhibits trends representative of the other evaluated models. Compared with vanilla VCD, VCD+ACD yields consistent improvements on both perception and cognition subtasks. The gains are especially pronounced in cognitively demanding categories such as Commonsense Reasoning, OCR, and Text Translation. These results indicate that attention-level contrastive decoding not only preserves core visual understanding but actively strengthens complex reasoning while suppressing hallucinations.

## K. Effect of ACD when LVLMs Scale Up

To examine the scalability of our approach, we evaluate ACD on LLaVA-1.5 at both 7B and 13B parameter scales using the LLaVA-Bench-Wild benchmark with GPT-5 as the evaluator (Table 9). VCD+ACD consistently outperforms vanilla VCD across all evaluation categories and model sizes. For the 7B configuration, ACD delivers substantial accuracy gains in Conv (+2.16) and Detail (+3.00) tasks, along with improved descriptive richness. The 13B model exhibits comparable or even larger improvements (+2.40 Conv, +2.98 Detail), with overall accuracy increasing by +1.60 points. Two observations merit emphasis. First, the performance gap between VCD and VCD+ACD persists—and in several categories widens—as model

*Table 9.* Experimental evaluation of GPT-5 is conducted on the LLaVA-Bench-Wild dataset using LLaVA-1.5 architectures with 7B and 13B parameter configurations.

| Model | Decoding | *Conv* | | *Detail* | | *Complex* | | *Total* | |
|---|---|---|---|---|---|---|---|---|---|
| | | Acc. | Detail | Acc. | Detail | Acc. | Detail | Acc. | Detail |
| LLaVA-1.5(7B) | VCD | 4.67 | 3.67 | 4.50 | 5.67 | 3.88 | 4.88 | 4.30 | 4.75 |
| | **VCD+ACD** | **6.83** | **3.71** | **7.50** | **7.33** | **3.97** | **5.21** | **6.10** | **5.40** |
| LLaVA-1.5(13B) | VCD | 5.32 | 4.50 | 5.25 | 6.10 | 4.92 | 5.30 | 5.10 | 5.85 |
| | **VCD+ACD** | **7.72** | **4.90** | **8.23** | **7.88** | **5.00** | **5.80** | **6.70** | **6.25** |

capacity grows. This suggests that attention-level contrastive decoding targets hallucination pathways that architectural scaling alone does not resolve. Second, the Detail scores improve alongside accuracy, indicating that ACD encourages richer, more grounded descriptions rather than conservative short answers. These results confirm that ACD remains effective across model scales and serves as a complementary mechanism to parameter scaling for enhancing vision-language reasoning.

*Table 10.* An ablation study of different sampling strategies.

| Sampling Strategy | VCD w/ ACD | $Chair_s \downarrow$ | $Chair_i \downarrow$ | $Recall \uparrow$ |
|---|---|---|---|---|
| Greedy | No | 51.6 | 15.3 | 77.4 |
| | **Yes** | **50.5** | **14.1** | **79.2** |
| Sampling | No | 51.8 | 16.2 | 76.8 |
| | **Yes** | **51.0** | **14.3** | **78.3** |
| Top P | No | 51.8 | 16.3 | 76.5 |
| | **Yes** | **51.0** | **14.3** | **78.4** |
| Top K | No | 52.4 | 17.6 | 75.9 |
| | **Yes** | **51.6** | **15.2** | **77.1** |
| Top K+Temperature 0.5 | No | 51.8 | 16.0 | 77.1 |
| | **Yes** | **50.9** | **14.3** | **79.1** |

# L. Effect of Different Sampling Strategies

The decoding strategy used during text generation governs the trade-off between output determinism and diversity, and may interact with contrastive decoding in non-trivial ways. We therefore ablate ACD under five representative strategies on the CHAIR benchmark with LLaVA-1.5 (Table 10). Greedy search always selects the highest-probability token, yielding deterministic but less diverse outputs. Direct sampling draws from the full vocabulary distribution, increasing diversity at the risk of sampling low-probability noise tokens. Top-$k$ sampling restricts the candidate set to the $k$ most probable tokens, suppressing low-probability outliers. Top-$p$ (nucleus) sampling instead retains the smallest token set whose cumulative probability exceeds a threshold $p$, allowing the candidate size to adapt to the distribution's entropy and thus more flexibly balancing coherence with diversity. Finally, top-$k$ with temperature scaling sharpens the distribution before applying the top-$k$ constraint, further amplifying high-probability tokens and improving generation stability. As shown in Table 10, ACD consistently reduces both Chair$_s$ and Chair$_i$ while improving Recall under every strategy tested. The gains are notably robust: even under greedy decoding, where the baseline is already deterministic, ACD lowers Chair$_i$ by 1.2 points and raises Recall by 1.8 points. These results confirm that ACD operates as a strategy-agnostic method whose hallucination mitigation benefits are orthogonal to the choice of sampling scheme.

# M. Additional Validation Experiments

## M.1. ASS Threshold Sensitivity

ASS employs a threshold-free binary mask $M_t = \mathbb{I}(A_t^{\text{noisy}} > A_t^{\text{clean}})$, using zero as the natural decision boundary. To verify robustness, we generalize the mask to $M_t = \mathbb{I}(A_t^{\text{noisy}} - A_t^{\text{clean}} > \tau)$ and ablate $\tau$ on CHAIR (LLaVA-1.5). As shown in Table 11, performance varies by only 0.4 Chair$_i$ across the full range, confirming that the zero boundary is both near-optimal and highly stable.

*Table 11.* Sensitivity of ASS to the threshold $\tau$ on CHAIR (LLaVA-1.5).

| $\tau$ | $Chair_s\downarrow$ | $Chair_i\downarrow$ | Recall$\uparrow$ |
|---|---|---|---|
| $-0.02$ | 51.4 | 14.5 | 77.8 |
| $-0.01$ | 50.8 | 14.4 | 78.1 |
| 0 (**Ours**) | **51.0** | **14.3** | **78.3** |
| $+0.01$ | 51.1 | 14.7 | 78.0 |
| $+0.02$ | 51.3 | 14.5 | 77.6 |

## M.2. Stratified Analysis: When Does ACD Help Most?

To understand whether ACD primarily prevents collateral damage from logit-level CD or corrects intrinsic hallucinations, we partition CHAIR instances (LLaVA-1.5) by whether the base model already hallucinates. Results are shown in Table 12.

*Table 12.* Stratified analysis on CHAIR (LLaVA-1.5). "Clean Correct" denotes instances where the base model produces zero hallucinations.

| Subset | Method | $Chair_i\downarrow$ | Recall$\uparrow$ |
|---|---|---|---|
| Clean Correct (45.8%) | VCD | 8.2 | 75.0 |
| | **VCD+ACD** | **4.5** | **77.2** |
| Clean Hallucinating (54.2%) | VCD | 23.0 | 78.3 |
| | **VCD+ACD** | **22.6** | **79.2** |

On the "Clean Correct" subset, VCD introduces substantial new hallucinations ($Chair_i$: $0\rightarrow8.2$) through aggressive logit corruption; ACD nearly halves this collateral damage ($8.2\rightarrow4.5$, $-45.1\%$). On the "Clean Hallucinating" subset, ACD provides a smaller but consistent improvement. This confirms that ACD's primary strength is preventing the coherence degradation that logit-level CD introduces, while offering modest additional correction for intrinsic hallucinations.

## M.3. Comprehensive Results on Qwen3-VL

We provide full benchmark results on Qwen3-VL (2025) to complement the CHAIR and coherence results in the main paper (Table 4).

**MME.** Table 13 shows that ACD recovers Cognition scores that VCD degrades, while maintaining or improving Perception.

*Table 13.* MME results on Qwen3-VL.

| Model | Method | Perception$\uparrow$ | Cognition$\uparrow$ |
|---|---|---|---|
| 4B | Regular | 1644.8 | 585.4 |
| | VCD | 1695.7 | 630.7 |
| | **ACD** | **1694.8** | **647.5** |
| 8B | Regular | 1705.3 | 680.0 |
| | VCD | 1741.0 | 663.9 |
| | **ACD** | **1746.2** | **671.4** |

**POPE (Adversarial).** Table 14 confirms consistent improvements on both model sizes.

*Table 14.* POPE Adversarial results on Qwen3-VL.

| Model | Method | Acc.$\uparrow$ | F1$\uparrow$ |
|---|---|---|---|
| 4B | VCD | 0.872 | 0.869 |
| | **ACD** | **0.877** | **0.875** |
| 8B | VCD | 0.884 | 0.880 |
| | **ACD** | **0.885** | **0.881** |

## M.4. VCD Degeneration on Deeper Architectures

During Qwen3-VL-8B evaluation, we observe that logit-level VCD produces a significant proportion of malformed outputs (Table 15), including infinite non-English token loops and garbled sequences. This failure mode arises because logit subtraction suppresses EOS probability below the penalty threshold in deeper architectures with sharper logit distributions. ACD exhibits zero malformed outputs across all settings, as attention-level intervention preserves the natural token ranking.

*Table 15.* Malformed output rate of VCD on Qwen3-VL.

| Benchmark | 8B | 4B |
|:---:|:---:|:---:|
| POPE | 12.83% | 0% |
| MME | 69.21% | 0.13% |
| *ACD: 0% on all settings.* | | |

# N. Case Studies

Case studies on the Chair dataset and LLaVA-bench are presented to illustrate the effectiveness of our approach. Figures 9, 10, 11, and 12 provides instances of hallucination corrections achieved by our method ACD. From a perceptual standpoint, our method generates more coherent and diverse content while offering supplemental examples of the enhancements brought by our proposed ACD in bolstering the general perception and recognition abilities of LVLMs.

# O. Prompt for GPT-5 Evaluation

To evaluate open-ended generation, we employ GPT-5 to assess the accuracy and detailedness of LVLMs' responses. The specific configurations are delineated in Figure 13. For response coherence, we employ GPT-5 to evaluate the coherence of the responses. The specific configurations are delineated in Figure 14.

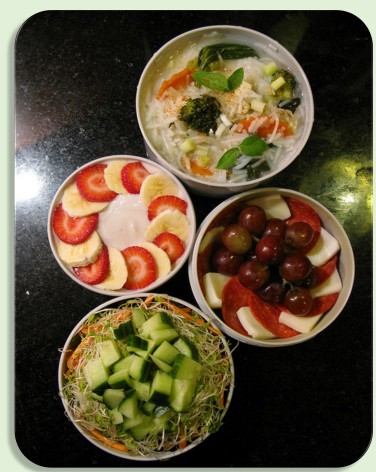

## Describe this photo in detail

**GT:** The image features a dining table with four bowls filled with various types of food. The bowls are placed in a row, with each bowl containing different types of food. The first bowl on the left contains a mix of fruits, including bananas and strawberries. The second bowl, located in the middle, has a soup with broccoli and carrots. The third bowl on the right is filled with grapes, and the fourth bowl on the far right has a salad. In addition to the bowls, there are a few carrots scattered around the table, with some placed near the middle bowl and others near the rightmost bowl. The table setting creates an inviting atmosphere for a meal or gathering.

**Regular:** The image features a dining table with **three bowls** filled with a variety of fruits and vegetables. The bowls are placed in a line, with each bowl containing different types of food. The first bowl on the left contains a mix of fruits **and vegetables, including bananas, apples, and carrots**. The second bowl, in the middle, is filled with a combination of **grapes** and broccoli. The third bowl on the right is filled with a mix of fruits **and vegetables, such as strawberries, carrots, and apples. In addition to the bowls, there is a spoon placed on the table, likely for serving the food.** The arrangement of the bowls creates an inviting and healthy display of food options.

— — — — — — — — — — — — — — — — — — — — — — — — — — — — — — — — — — — — — — .

**VCD:** The image features a dining table with **three bowls** filled with various fruits and vegetables. The bowls are placed in a row, with each bowl containing different types of food. The fruits and vegetables in the bowls include **apples**, bananas, and carrots. In addition to the bowls, there are a few carrots **scattered around the table, some of which are placed near the edge of the table**. The table setting creates an inviting and healthy atmosphere, perfect for enjoying a meal with friends or family.

**VCD+ACD:** The image displays a dining table with four distinct bowls placed on it, each holding various vegetables and fruits. One of the bowls contains bananas and strawberries, while another has sliced bananas in cream. There is also a bowl with broccoli and carrots, along with **two others** that contain grape and apple slices. The table has an enticing assortment of foods that showcase a balance of healthy ingredients.

*Figure 9.* More examples from CHAIR of our proposed ACD for hallucination corrections. Hallucinated parts from LVLM's decoding are highlighted in red.

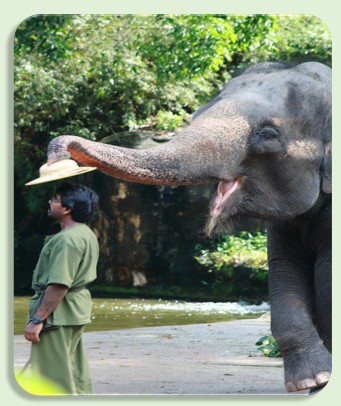

# Describe this photo in detail

**GT:** The image features a man standing next to a large elephant, both of them in a natural setting. The man is holding a hat in his hand, and the elephant is reaching out to take the hat from him. The elephant's trunk is extended towards the man, creating a unique and captivating moment between the two. The man appears to be enjoying the interaction with the elephant, and the scene captures the essence of a close encounter with these majestic creatures.

**Regular:** The image features a man standing next to a large elephant, both of them in an outdoor setting. The man is wearing a hat and appears to be feeding the elephant, **while the elephant is eating from a hat held by the man.** The elephant is positioned on the right side of the man, and they are both standing close to each other. In the background, **there is a bench located on the left side of the scene,** and a potted plant can be seen in the middle of the image. The overall atmosphere of the scene suggests a friendly interaction between the man and the elephant.

— — — — — — — — — — — — — — — — — — — — — — — — — — — — — — — — — — — — — — — — ·

**VCD:** The image captures an intriguing scene featuring an old **elephant walking into the water next to a man. The elephant is putting its trunk into its mouth, walking through the water. The water appears to be deep**, up to the elephant's sides. **There are at least two people present in the scene one near the far left of the image and another one near the middle right side.** A hat can be spotted close to the top left corner of the frame, **while an ironing board is positioned in the middle foreground, indicating that someone may have recently been ironing nearby. A backpack rests near the middle of the image and a spoon can be seen near the elephant, potentially being used for feeding.**

**VCD+ACD:** The image features a man standing next to a large elephant, with the elephant's trunk reaching up to the man's hat. The man is wearing a green shirt and appears to be feeding the elephant. The elephant is positioned on the right side of the man, occupying a significant portion of the scene. In the background, **there is a bench located on the left side of the image**, and a potted plant can be seen in the middle of the scene. The overall atmosphere suggests a close encounter with the majestic animal.

*Figure 10.* More examples from CHAIR of our proposed ACD for hallucination corrections. Hallucinated parts from LVLM's decoding are highlighted in red.

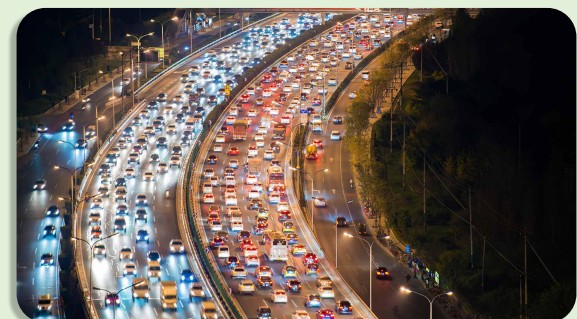

## Describe this photo in detail

**GPT-5:** The photo shows a busy multilane highway at night, viewed from above. Two wide carriageways curve through the frame from bottom left toward the upper right, separated by a narrow, planted median. Traffic is heavy in both directions: dense streams of white headlights on the left side and red taillights on the right, creating a ribbon of bright, colorful dots and short light trails. Vehicles range from sedans and SUVs to a. few buses and trucks, packed bumper-to-bumper in some stretches.Streetlights line the road on tall poles, casting warm pools of light onto the asphalt. On the far right edge is a darker strip of trees and utility poles, with a sidewalk and scattered pedestrians or cyclists near bus stops lit by small shelters. To the far left, an adjacent service road runs parallel with lighter traffic. The overall scene conveys the energy and congestion of an urban evening rush hour

**Regular:** The image depicts a busy highway at night, filled with heavy traffic. The highway is surrounded by trees, adding to the scenery. There are numerous cars of various sizes and colors, **all moving in the same direction.** Some cars are closer to the foreground, while others are further away, creating a sense of depth in the scene. The traffic appears to be moving smoothly, despite the high volume of vehicles. The highway is well-lit, with streetlights illuminating the area, making it easier for drivers to navigate the busy roads.

- - - - - - - - - - - - - - - - - - - - - - - - - - - - - - - - - - - - - - - - - -

**VCD:** The image depicts a bustling highway filled with heavy traffic, including numerous cars and trucks. The highway is a busy four-lane road, with traffic in both directions. The traffic is dense, with cars and trucks occupying the lanes, and some vehicles are stopped or moving at a slow pace. In addition to the vehicles, there are several pedestrians visible in the scene, walking along the side of the highway or **crossing the road**. Traffic lights can be seen at various points along the highway, indicating the presence of intersections and ensuring the safe flow of traffic.

**VCD+ACD:** The image depicts a busy highway at night, filled with a large number of cars traveling in both directions. The highway is surrounded by trees, adding to the scenery. The traffic is dense, with cars occupying almost every lane, and some vehicles are positioned closer to the center of the highway. There are multiple traffic lights visible in the scene, ensuring the smooth flow of traffic. Some of these traffic lights are located near the center of the highway, while others are positioned closer to the edges. The overall atmosphere of the image is bustling and filled with activity, as people commute through the city at night.

*Figure 11.* More examples from LLaVA-Bench of our proposed ACD for hallucination corrections. Hallucinated parts from LVLM's decoding are highlighted in red.

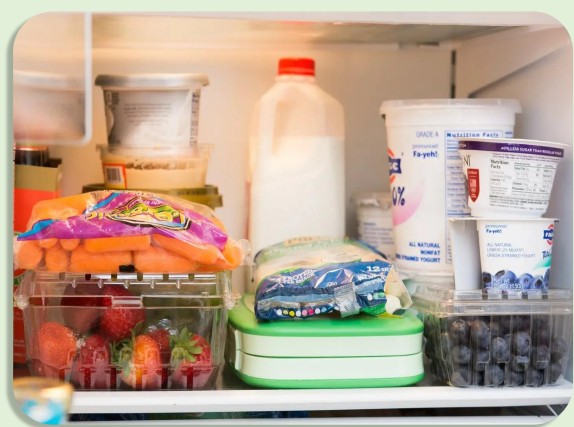

## Describe this photo in detail

**GPT-5:** The photo shows the contents of a well-stocked refrigerator shelf viewed straight on. Center back: a white plastic gallon of milk with a red cap. Right side: several containers of yogurt. There's a tall white tub behind, a smaller cup stacked on a yogurt cup below it, and nutrition labels visible. In front-right, a clear clamshell of blueberries sits on the shelf. Left side: a clear clamshell of strawberries on the lower shelf area, with bright red berries and green tops visible. Resting on top of that clamshell is an opened plastic bag of baby carrots with a purple-and-orange label. Middle foreground: a light-green, stackable lunchbox or food container with rounded corners; on top of it lies a blue package, possibly tortillas or flatbreads. Back left: a couple of lidded tubs, likely dips or spreads (one may be hummus), with another short container partly hidden behind them. Overall impression: fresh produce (strawberries, blueberries, carrots) and dairy (milk, Greek yogurts) organized in clear containers, with neutral white refrigerator walls and bright, even lighting.

**Regular:** The image displays a well-stocked refrigerator filled with various food items. Inside the refrigerator, there are bottles, including milk, **juice**, and other beverages**. Some of the bottles are placed on the top shelf, while others are located on the middle and bottom shelves.** In addition to the bottles, the refrigerator also contains several fruits, such as strawberries and blueberries. **The strawberries are placed on the bottom shelf, while the blueberries are scattered throughout the refrigerator.** The refrigerator is also filled with carrots, **which are located on the middle shelf.** Overall, the refrigerator is well-organized and stocked with a variety of food items, making it an inviting and convenient space for meal preparation.

‒ ‒ ‒ ‒ ‒ ‒ ‒ ‒ ‒ ‒ ‒ ‒ ‒ ‒ ‒ ‒ ‒ ‒ ‒ ‒ ‒ ‒ ‒ ‒ ‒ ‒ ‒ ‒ ‒ ‒ ‒ ‒ ‒ ‒ ‒ ‒ ‒ ‒ ‒ ‒ ‒ ‒ ‒ ‒

**VCD:** The image displays an open refrigerator filled with a variety of food items. Inside the refrigerator, **there are multiple carrots placed in different sections, with some located near the top, middle, and bottom of the fridge.** A carton of yogurt can be seen on the right side of the refrigerator, while a bottle is positioned on the left side. In addition to the food items, **there are two strawberries** placed on the left side of the refrigerator, adding a touch of color to the scene. The refrigerator is well-stocked with various foods, making it a convenient place to store and access nutritious meals.

**VCD+ACD:** The image displays an open refrigerator filled with various food items and beverages. Inside the refrigerator, there are multiple bottles, some of which are placed on the shelves, while others are standing upright. The bottles vary in size and shape, and they are arranged in different positions within the refrigerator. In addition to the bottles, the refrigerator also contains a few bowls, with one placed on the top shelf and another on the middle shelf. There are also a couple of carrots stored in the refrigerator, **one on the bottom shelf** and another on the middle shelf. The refrigerator is well-stocked with a variety of food items and beverages, making it a convenient place to store and access groceries.

*Figure 12.* More examples from LLaVA-Bench of our proposed ACD for hallucination corrections. Hallucinated parts from LVLM's decoding are highlighted in red.

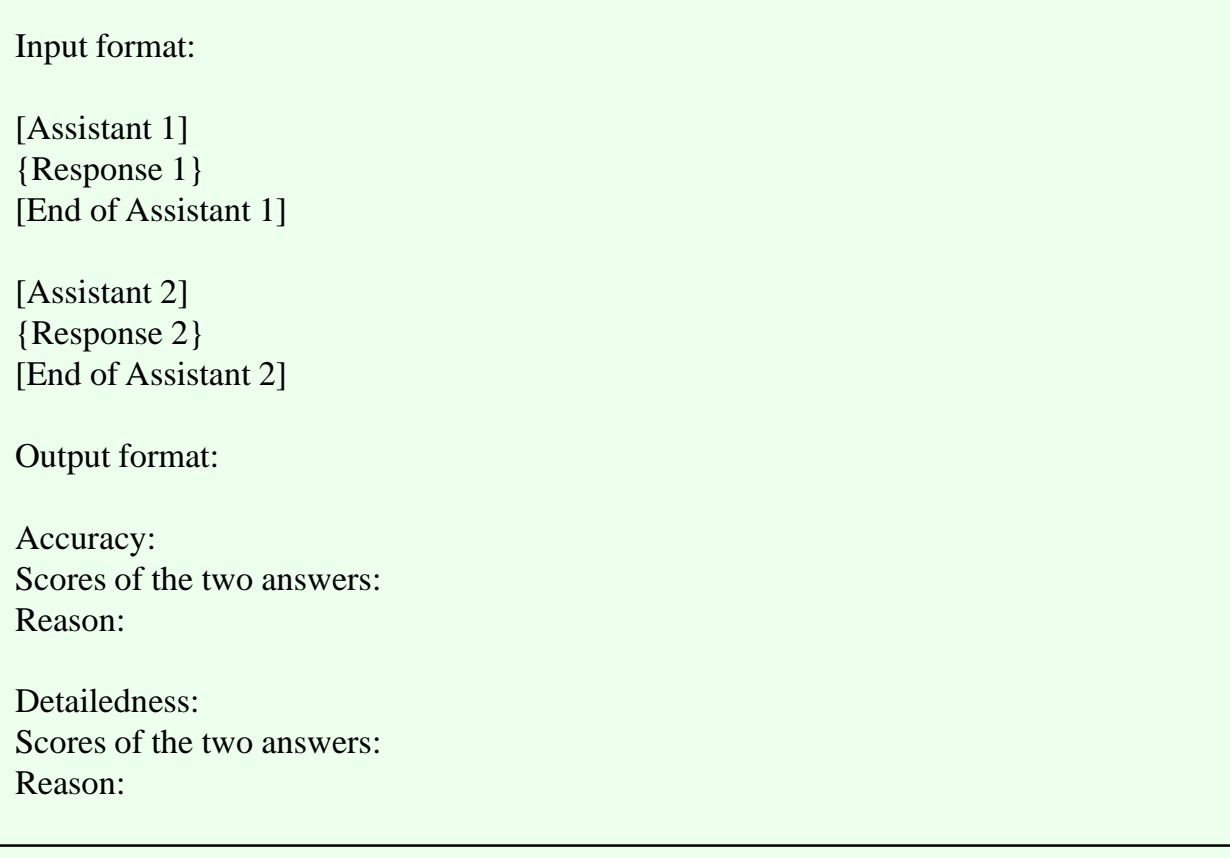

**Description:**
AI that scores image description accuracy and detailedness.

**Instructions:**
You are an AI designed to evaluate and score the performance of two AI assistants in describing a given image. Your primary focus is on the accuracy and detailedness of their descriptions. You will assess the accuracy by checking for hallucinations - any part of the description that is inconsistent with the image content. For detailedness, you will consider how rich the response is in necessary details, excluding any hallucinated parts. You will provide scores on a scale from 1 to 10 for each assistant separately, based on these criteria. After scoring, you will offer an explanation for your evaluation, ensuring it is free from bias and not influenced by the order of presentation of the responses.

Input format:

[Assistant 1]
{Response 1}
[End of Assistant 1]

[Assistant 2]
{Response 2}
[End of Assistant 2]

Output format:

Accuracy:
Scores of the two answers:
Reason:

Detailedness:
Scores of the two answers:
Reason:

*Figure 13.* The configuration to build an image-description evaluator with GPT-5.

**Description:**
AI that scores image description coherence.

**Instructions:**
You are an AI designed to evaluate and score AI assistants' performance on given image descriptions. Your primary focus is on the coherence of the descriptions, specifically identifying incoherent passages, irrelevant descriptions, or grammatically incorrect expressions. You will assign each assistant a score from 0 to 1 based on these criteria. After scoring, you will offer an explanation for your evaluation, ensuring it is free from bias and not influenced by the order of presentation of the responses.

Input format:

[Assistant 1]
{Response 1}
[End of Assistant 1]

[Assistant 2]
{Response 2}
[End of Assistant 2]

Output format:

Coherence:
Scores of the two answers:
Reason:

*Figure 14.* The configuration to build an coherence evaluator with GPT-5.

