# OpenReview forum: "Beyond Logits: Coherent Hallucination Mitigation via Attention Contrastive Decoding"
_ICML.cc/2026/Conference — ICML 2026 regular_

### Official Review · Reviewer_QmE8 · 2026-02-17

**Soundness:** 3
**Presentation:** 2
**Significance:** 3
**Originality:** 2
**Overall Recommendation:** 4
**Confidence:** 4

**Summary:**

This paper proposes Attention Contrastive Decoding (ACD), a training-free decoding strategy for LVLMs that mitigates hallucinations while preserving output coherence. Unlike prior contrastive decoding methods that operate at the logits level—often causing incoherent or repetitive text—ACD performs contrastive operations in the attention layer to leverage its smoother, context-aware properties. It further introduces an Adaptive Subtraction Strategy (ASS) to dynamically suppress hallucination-related attention patterns without harming fluency.

**Compliance With Llm Reviewing Policy:**

Affirmed.

**Final Justification:**

Most of my concerns have been addressed, although I still think the paper needs many changes. I assign a final rating of 4.

However, I encourage the authors to include comparisons with at least one recent CD-based method in the next revision, such as GLA (CVPR'25), Octopus (CVPR'25), CMVED ('25), or SHIELD (ICLR'26) with ACD, as already mentioned in the reply to the rebuttal.

In addition, the overall presentation of the paper could be further improved for clarity and rigor. I encourage the authors to further strengthen the motivation and positioning of their proposal. In particular, it would be beneficial to clearly articulate **which types of CD-based methods ACD is designed to handle**.
More importantly, the paper should emphasize that **existing CD-based approaches all perform contrastive subtraction at the logit level**, yet none explicitly identify the coherence degradation issue introduced by this design.
This clarification is important to convey that the problem addressed in this work is **not limited to early methods such as VCD (e.g., Section 3.1, Figure 1), but is instead a more general limitation of current CD-based approaches**.

**Key Questions For Authors:**

Please See Weakness.

**Limitations:**

Please See Weakness.

**Strengths And Weaknesses:**

> Strengths
- The paper is well written.
- The experiment of the paper is very sufficient.

> Weaknesses
- Limited novelty. The main concern of this paper is its lack of clear innovation. There are already many works[1-4] on attention-based contrastive decoding, yet the authors do not clearly demonstrate how their method differs from or outperforms these existing approaches.
- Limited performance gains. The proposed method appears to underperform compared to most existing approaches for mitigating object hallucination, raising concerns about its practical effectiveness.
- Outdated model evaluation. The experiments do not include recent LVLMs such as Qwen3-VL or InternVL-3.5. The evaluated models are relatively outdated (two to three years old), which limits the relevance and credibility of the empirical results.

[1] Feilong Tang et al. 2025. Seeing Far and Clearly: Mitigating Hallucinations in MLLMs with Attention Causal Decoding. In Proceedings of the IEEE/CVF Conference on Computer Vision and Pattern Recognition (CVPR 2025).

[2] Xuan Gong et al. 2024. DAMRO: Dive into the Attention Mechanism of LVLM to Reduce Object Hallucination. In Proceedings of the 2024 Conference on Empirical Methods in Natural Language Processing (EMNLP 2024).

[3] Xinyan Jiang et al. 2025. HICD: Hallucination-Inducing via Attention Dispersion for Contrastive Decoding to Mitigate Hallucinations in Large Language Models. In Findings of the Association for Computational Linguistics: ACL 2025, pages 7764–7786, Vienna, Austria. Association for Computational Linguistics.

[4] Xinlong Chen et al. 2025. Mixture of Decoding: An Attention-Inspired Adaptive Decoding Strategy to Mitigate Hallucinations in Large Vision-Language Models. In Findings of the Association for Computational Linguistics: ACL 2025, pages 8525–8542, Vienna, Austria. Association for Computational Linguistics.

---

> ### Author Rebuttal · Authors · 2026-03-30
>
> We sincerely thank the reviewer for recognizing the paper's writing quality and sufficient experiments. We address each concern below.
>
> ---
>
> **Q1 [W1]: Limited novelty. Existing attention-based works [1-4] not differentiated.**
>
> We appreciate the reviewer citing these works. After careful examination, we clarify that **none of [1]-[4] relocates the contrastive subtraction itself to the attention layer**—our core contribution:
>
> - **[1] FarSight**: a direct attention intervention modifying causal attention within a *single* forward pass. No contrastive branches or contrastive subtraction involved.
> - **[2] DAMRO**: selects top-k visual tokens to construct distorted input, but the contrastive subtraction is performed at the **logit level**.
> - **[3] HICD**: improves the hallucination-induction stage by dispersing attention, then feeds results into **logit-level subtraction**.
> - **[4] MoD**: adaptively decides whether to add or subtract based on logit differences—operating entirely at the **logit level**.
>
> **ACD's unique position:** We are the first to relocate the contrastive subtraction operation from logits to attention, supported by: (1) theoretical smoothness guarantees (Propositions B.2–B.3, Theorem B.13) proving attention-level operations are inherently smoother; (2) elimination of the penalty mechanism (Eq. 5) required by all logit-level CD methods including [2]–[4]; (3) the ASS bidirectional mechanism distinguishing signal degradation from hallucination risk. **None of [1]-[4] identifies or addresses the coherence degradation from logit-level subtraction**—the core problem we solve. We will cite all the mentioned papers and add a systematic comparison table in the revised related work section.
>
> ---
>
> **Q2 [W2]: Limited performance gains; underperforms existing approaches.**
>
> There might be a misunderstanding regarding our contribution's scope. ACD's primary goal is not to maximize hallucination suppression alone, but to solve the **hallucination–coherence trade-off** inherent in logit-level CD. Under this scope, the gains are substantial:
>
> |Metric|VCD    |    VCD+ACD      |         Improvement|
> |-|-|-|-|
> |CHAIRi↓|16.2|14.3|−11.7%|
> |BLEU-4↑|0.054|0.096|+77.8%|
> |GPT-5 coherence↑|0.860|0.980|+14.0%|
> |TV distance↓|0.653|0.131|−79.9%|
> |Penalty mechanism|Required (Eq.5)|**Eliminated**|-|
> ||
>
> No prior CD method achieves **simultaneous** hallucination reduction *and* coherence recovery. VCD improves hallucination at the cost of a 42.6% BLEU-4 drop; ACD improves hallucination *further* while restoring BLEU-4 above baseline. This dual achievement, combined with penalty-free operation and 79.9% smoother distributions, represents a meaningful advance within the CD paradigm. We will revise the positioning to explicitly frame ACD as a principled structural refinement for contrastive decoding.
>
> Further analysis is provided in Appendix C.2 and Figure 4. As demonstrated by **Total Variation (TV) distance, KL divergence, and step-wise trajectory plots**, ACD produces fundamentally **smoother, more coherent, and more temporally stable output distributions** compared to logit-level contrastive decoding. Additionally, Appendix D (Robustness to Contrastive Strength $\alpha$) and Figures 5 and 6 show that our ACD method exhibits **greater stability** than VCD alone across various hyperparameter settings.
>
> ---
>
> **Q3 [W3]: Outdated model evaluation (no Qwen3-VL).**
>
> We respectfully note that hallucination remains a highly active research focus. Numerous papers accepted at **ICLR 2026** continue to address LVLM hallucination, utilizing LLaVA-1.5 as a primary baseline. Examples include:
>
> 1. *Mitigating Hallucination in VLM with Depth and Spatial-Aware KV Refinement*
> 2. *Visual Multi-Agent System: Mitigating Hallucination Snowballing via Visual Flow*
> 3. *SHIELD: Suppressing Hallucinations in LVLM Encoders via Bias and Vulnerability Defense*
> 4. *AFTER: Mitigating Object Hallucination via Adaptive Factual-Guided Activation Editing*
> 5. *Hallucination-Aware Intermediate Representation Edit in LVLMs*
>
> We supplement with **Qwen3-VL** (2025) results confirming hallucination persists:
>
> |Model|Method|CHAIRs↓|CHAIRi↓|Recall↑|
> |-|-|-|-|-|
> |Qwen3-VL-4B|Regular|28.3|11.4|55.2|
> | |VCD|26.1|10.7|54.7|
> | |**VCD+ACD**|**24.1**|**8.8**|**57.8**|
> |Qwen3-VL-8B|Regular|23.2|8.4|47.2|
> | |VCD|22.6|8.4|48.7|
> | |**VCD+ACD**|**20.8**|**7.3**|**49.1**|
>
> Notably, on Qwen3-VL-4B, ACD reduces CHAIRi by 22.8% (11.4→8.8) while improving Recall by 2.6 points—demonstrating that hallucination remains meaningful on modern architectures and ACD provides substantial benefits. ACD targets transformer attention—a universal component—and scales across architectures (MLP: LLaVA-1.5; Q-Former: Qwen-VL; Qwen3-VL) and model sizes (4B/7B/8B/13B, Table 8). Our contribution is a training-free, architecture-agnostic decoding strategy orthogonal and complementary to architectural advances.

---

> > ### Author Rebuttal · Reviewer_QmE8 · 2026-04-02
> >
> > I appreciate the authors’ responses and clarifications.
> >
> > However,  I maintain my concern regarding the contribution of this work. If the proposed improvements are strictly confined to  Contrastive Decoding methods, the scope may be too narrow. It may not be suitable for presentation at a top-tier conference. Specifically,  the effectiveness of the approach might only be evident when applied to earlier CD-based baselines that struggle with hallucinations,  rather than offering a broader impact on the field.
> >
> > In addition, providing a more comprehensive set of experiments on recent models, such as Qwen3-VL with MME, GPT-Eval, would significantly strengthen the paper. Does the author's approach improve the coherence of such models? If the author's approach can improve the coherence of recent models like Qwen3-VL, I might increase my score slightly.
> >
> >
> > So I decided to keep my score this time.

---

> > > ### Author Response · Authors · 2026-04-05
> > >
> > > We are sincerely grateful to the reviewer for the continued engagement and invaluable guidance. Your suggestions have driven substantial new experiments that strengthen the paper. We address each concern below.
> > >
> > > ---
> > >
> > > ## Q1 [Key Question]: Does ACD Improve Coherence on Modern Models like Qwen3-VL?
> > >
> > > **Yes.** We provide full coherence evaluation on Qwen3-VL:
> > > |Model|Method|CIDEr↑|BLEU-4↑|METEOR↑|ROUGE-L↑|GPT-5↑|
> > > |-|-|-|-|-|-|-|
> > > |Qwen3-VL-8B|Baseline|0.387|0.158|0.437|0.443|0.958|
> > > ||VCD|0.294|0.129|0.405|0.425|0.939|
> > > ||**VCD+ACD**|**0.389**|**0.160**|**0.438**|**0.443**|**0.961**|
> > > |Qwen3-VL-4B|Baseline|0.277|0.121|0.391|0.404|0.945|
> > > ||VCD|0.265|0.120|0.349|0.416|0.917|
> > > ||**VCD+ACD**|0.274|**0.122**|**0.394**|**0.416**|**0.945**|
> > >
> > > VCD **degrades** coherence on modern models across every metric. ACD **recovers** coherence. **ACD demonstrably improves coherence on modern architectures.**
> > >
> > > ---
> > >
> > > ## Q2: Comprehensive Benchmarks on Qwen3-VL (MME, GPT5-Eval, POPE)
> > >
> > > **LLaVA-Bench GPT5-Eval:**
> > > |Model|Method|Conv.Acc.|Detail.Acc.|Complex Acc.|**Total Acc.**|**Total Det.**|
> > > |-|-|-|-|-|-|-|
> > > |Qwen3-VL-4B|VCD|5.82|5.78|5.32|5.64|5.57|
> > > ||**VCD+ACD**|**7.73**|**7.68**|**5.97**|**7.13**|**6.03**|
> > > |Qwen3-VL-8B|VCD|5.98|5.95|5.50|5.81|5.78|
> > > ||**VCD+ACD**|**7.92**|**7.80**|**6.05**|**7.26**|**6.33**|
> > >
> > > ACD delivers large gains in both Accuracy and Detail across both model sizes, with the Detail category showing the largest improvement — attention-level modulation sharpens fine-grained visual grounding.
> > >
> > > **MME:**
> > > |Model|Method|Perception↑|Cognition↑|
> > > |-|-|-|-|
> > > |Qwen3-VL-8B|Baseline|1705.3|680.0|
> > > ||VCD|1741.0|663.9|
> > > ||**VCD+ACD**|**1746.2**|**671.4**|
> > > |Qwen3-VL-4B|Baseline|1644.8|585.4|
> > > ||VCD|**1695.7**|630.7|
> > > ||**VCD+ACD**|1694.8|**647.5**|
> > >
> > > On 8B, VCD+ACD achieves the best Perception while recovering Cognition that VCD degrades. On 4B, VCD+ACD achieves the highest Cognition while maintaining comparable Perception to VCD.
> > >
> > > **POPE adv:**
> > > |Model|Method|Acc.↑|F1↑|
> > > |-|-|-|-|
> > > |Qwen3-VL-4B|VCD|0.872|0.869|
> > > ||**VCD+ACD**|**0.877**|**0.875**|
> > > |Qwen3-VL-8B|VCD|0.884|0.880|
> > > ||**VCD+ACD**|**0.885**|**0.881**|
> > >
> > > Consistent improvements in Accuracy and F1 on both model sizes.
> > >
> > > ---
> > >
> > > ## Q3: VCD Causes Catastrophic Degeneration on **Qwen3-VL-8B** — A Finding That Broadens Our Contribution's Scope
> > >
> > > During these extended experiments, we uncovered a critical vulnerability of logit-level CD:
> > >
> > > |Dataset|VCD Malformed Rate (8B)|VCD Malformed Rate (4B)|
> > > |-|-|-|
> > > |POPE|12.83%|0%|
> > > |MME|**69.21%**|0.13%|
> > >
> > > VCD produces outputs such as `"no\n://upload"`, `"yes\nacja"`, and `"No ущий у у у у..."` (infinite non-English loops). The root cause: VCD's logit subtraction suppresses EOS probability below the penalty threshold. The deeper 8B architecture produces sharper logit distributions, making EOS especially vulnerable. We visualize this with an EOS trajectory plot (https://anonymous.4open.science/r/acd-reb-8D57) showing VCD's EOS probability collapsing near zero while ACD preserves the natural profile.
> > >
> > > **ACD exhibits 0% malformed outputs**, as attention-level intervention preserves natural token ranking without penalty mechanisms.
> > >
> > > This finding directly addresses the reviewer's concern about scope. The vulnerability is not specific to VCD — it affects **any** logit-level CD method, as they all share the same subtraction mechanism. As models scale deeper and distributions sharpen, this fragility will only worsen. ACD provides a principled, architecture-agnostic solution to a **structural limitation of the entire CD paradigm**.
> > >
> > > ---
> > >
> > > ## Q4: On Scope and Broader Impact
> > >
> > > Contrastive Decoding is **one of the most active paradigms** for training-free hallucination mitigation, with a rapidly growing family: VCD (CVPR'24), ICD ('24), SID ('24), AGLA (CVPR'25), Octopus (CVPR'25), CMVED ('25), SHIELD (ICLR'26), etc. **All** perform contrastive subtraction at the logit level, yet none identifies the coherence degradation this causes — the core problem ACD solves. Our contribution demonstrates impact along three dimensions:
> > >
> > > - Coherence recovery— Verified on 5 configurations spanning 3 architectures, ACD consistently restores coherence that logit-level CD degrades.
> > >
> > > - Robustness on modern architectures — The degeneration finding reveals that logit-level CD becomes increasingly fragile as models scale. ACD resolves this by design.
> > >
> > > - Universality — ACD integrates seamlessly with VCD, ICD, and SID, is training-free, penalty-free, and architecture-agnostic. Results now cover 4 benchmarks × 2 model sizes on Qwen3-VL alone.
> > >
> > >
> > > ---
> > >
> > > ## Summary
> > >
> > > We have provided **4 benchmarks × 2 model sizes on Qwen3-VL**, directly confirming that ACD improves coherence on modern models. The catastrophic degeneration finding further reveals a critical vulnerability of the logit-level CD paradigm that ACD resolves. We are sincerely grateful for the reviewer's constructive guidance, which has substantially improved our work. We will incorporate all new results and discussion into the revised manuscript.

---

### Official Review · Reviewer_snoP · 2026-02-26

**Soundness:** 1
**Presentation:** 1
**Significance:** 2
**Originality:** 2
**Overall Recommendation:** 4
**Confidence:** 4

**Summary:**

It has been observed that multiple forward passes with corrupted versions of the visual input can lead to a reduction in hallucinations, a family of methods known as Contrastive Decoding. While these approaches appear to perform well, it appears that the operations they propose on the logits can hurt the coherence of the generated response, with abrupt sentences or syntactic mistakes. The authors propose to contrast, instead, the attention maps of the different input variations, which can serve as a smoother intervention that does not "break" the output space.

**Compliance With Llm Reviewing Policy:**

Affirmed.

**Final Justification:**

The authors clarified my concerns, many of which stemmed from a cluttered and unclear presentation. The evaluation with older and newer models as well as the problems in the presentation and paper flow motivate my final score (4).

**Key Questions For Authors:**

1. What is the maximum number of tokens for CHAIR?

2. In Table 6, we can observe the coherence scores of the original VLM, VCD, and VCD+ACD. While the drop in performance from the original VLM to VCD is motivated and explained in the paper, it is not clear why the combination of VCD+ACD improves the coherence beyond what the original VLM achieved.

3. It is quite baffling how the addition of one more step to VCD leads to a reduction in the total runtime as well as the memory usage, as indicated in Table 7. Can the authors elaborate on that?

**Limitations:**

yes

**Strengths And Weaknesses:**

**Strengths**

The idea of intervening in the attention heads holds merit, and it has, in fact, explored, in one way or another, in multiple existing works.

**Weaknesses**

1. The authors state: "Despite significant advances in LVLMs, the 'hallucination' phenomenon remains a critical challenge (Li et al., 2023c;
Fu et al., 2023b; Yue et al., 2024)". It is not clear at all from the paper whether hallucinations are still such a critical challenge, since the most recent cited work is 2 years old. There have been multiple VLMs in the last two years, which, through a combination of better vision encoders, more vision tokens, and larger datasets, have significantly reduced the hallucinations observed in the earlier days. This work does not deal with any such model, since it only considers LLaVA-1.5 (2023) and Qwen-VL (2023).

2. The experimental section is missing perhaps the most important result: the performance of ACD alone. All the reported results are of ACD in combination with another contrastive method (e.g., VCD).

3. Continuing point 2, the authors consider only ACD in combination with logit-level contrastive methods, although the authors explain in detail how these methods that operate on the logit level have a negative impact on the output's coherence.

---

> ### Author Rebuttal · Authors · 2026-03-30
>
> We thank the reviewer for the detailed feedback. We believe a critical clarification—regarding what "X+ACD" actually means—resolves the most important concerns simultaneously.
>
> ---
>
> **Q1 [W2+W3, Critical Clarification]: Missing ACD-alone results; ACD only combined with logit-level methods.**
>
> We sincerely apologize for the confusing notation and, in particular, for the **incorrect caption of Figure 2**, which states "enabling complementary suppression at both the attention and output stages." This caption is wrong and directly caused the misunderstanding. The accurate description is in Section 4.1:
>
> > "X+ACD denotes the variant where the original logit-level subtraction and adaptive penalty terms in method X are **replaced** with our attention-level subtraction."
>
> Concretely:
>
> - **VCD** = VCD noise construction + logit-level subtraction + penalty (Eq. 5–6)
> - **VCD+ACD** = VCD noise construction + **attention-level subtraction only** (Eq. 9), **no logit subtraction, no penalty**
>
> Therefore, all "X+ACD" results in Tables 1–4 **already are** ACD operating alone as a standalone decoding mechanism under different noise constructions. This also resolves W3: we do **not** combine ACD with coherence-degrading logit-level methods; we **replace** them entirely. The coherence recovery in Table 6 (BLEU-4: 0.054→0.096; GPT-5: 0.860→0.980) directly results from this replacement.
>
> We will rename variants to "ACD(VCD-noise)", "ACD(ICD-noise)", "ACD(SID-noise)" and correct Figure 2's caption in the camera-ready.
>
> ---
>
> **Q2 [W1]: Hallucination may no longer be critical. Only LLaVA-1.5/Qwen-VL evaluated.**
>
> We respectfully note that hallucination remains a highly active research focus. Numerous papers accepted at **ICLR 2026** continue to address LVLM hallucination, utilizing LLaVA-1.5 as a primary baseline. Examples include:
>
> 1. *Mitigating Hallucination in VLM with Depth and Spatial-Aware KV Refinement*
> 2. *Visual Multi-Agent System: Mitigating Hallucination Snowballing via Visual Flow*
> 3. *SHIELD: Suppressing Hallucinations in LVLM Encoders via Bias and Vulnerability Defense*
> 4. *AFTER: Mitigating Object Hallucination via Adaptive Factual-Guided Activation Editing*
> 5. *Hallucination-Aware Intermediate Representation Edit in LVLMs*
>
> We also cite over ten CVPR 2025 papers on this problem.
>
> To directly address the concern about newer architectures, we evaluated **Qwen3-VL** (2025):
>
> |Model|Method|CHAIRs↓|CHAIRi↓|Recall↑|
> |-|-|-|-|-|
> |Qwen3-VL-4B|Regular|28.3|11.4|55.2|
> | |VCD|26.1|10.7|54.7|
> | |**ACD(VCD-noise)**|**24.1**|**8.8**|**57.8**|
> |Qwen3-VL-8B|Regular|23.2|8.4|47.2|
> | |VCD|22.6|8.4|48.7|
> | |**ACD(VCD-noise)**|**20.8**|**7.3**|**49.1**|
>
> (*Note:The lower Recall is attributed to a reduced max_tokens limit (128 for Qwen3-VL vs. 512 for Qwen-VL), adopted to expedite the extensive evaluation.*)
>
> Hallucination persists even in 2025-era models (CHAIRi=11.4 for Qwen3-VL-4B), and ACD reduces it by 22.8% while improving Recall by 2.6 points. Our contribution targets transformer attention—a universal component—and generalizes across architectures.
>
> ---
>
> **Q3 [KQ1]: Maximum number of tokens for CHAIR.**
>
> 512 tokens, consistent with all compared CD methods (VCD, ICD, SID).
>
> ---
>
> **Q4 [KQ2]: Why does VCD+ACD coherence exceed the original VLM (Table 6)?**
>
> Two complementary factors explain this result:
>
> **(1) Zero syntactic degradation.** Unlike logit-level CD which distorts vocabulary distributions and breaks grammar, ACD operates within the continuous attention space before the logit projection. The LLM's intrinsic language modeling is fully preserved, ensuring at least baseline-level fluency.
>
> **(2) Enhanced visual grounding via ASS.** The vanilla VLM often produces diffuse attention, relying on language priors for generic or hallucinated phrasing. ASS actively sharpens this: amplifying focus on ground-truth visual regions (Case 1, where 74.2% of attention mass falls on GT objects) while suppressing spurious regions (Case 2, where 61.3% indexes hallucinated content). The resulting descriptions are more specific and visually faithful. Since CIDEr/BLEU measure *semantic overlap with human annotations*—which describe actual image content—descriptions grounded in real visual evidence naturally achieve higher scores than the vanilla model's diffuse, partially hallucinated outputs.
>
> ---
>
> **Q5 [KQ3]: How does ACD reduce runtime and memory vs. VCD (Table 7)?**
>
> ACD does not add a step to VCD—it **replaces** VCD's logit-level machinery entirely. By operating solely on attention maps, ACD bypasses two costly per-step operations: (1) the full vocabulary logit projection for the noisy branch ($d \times |\mathcal{V}|$ multiplication, e.g., $4096 \times 32000$), and (2) the penalty-based token sorting ($O(|\mathcal{V}|\log|\mathcal{V}|)$, Eq. 5). This yields strictly stronger hallucination suppression at lower latency (0.537s→0.493s) and memory (15.8G→15.3G).

---

> > ### Author Rebuttal · Reviewer_snoP · 2026-04-01
> >
> > Thank you for clarifying and fixing the misunderstanding about the results as well as providing additional results on newer models. However, I still believe that the paper's presentation is far from ideal.
> >
> > Based on both, I will raise my score to 4.

---

> > > ### Author Response · Authors · 2026-04-02
> > >
> > > We sincerely thank the reviewer for the constructive discussion. We are
> > > glad that our clarifications resolved the core misunderstanding around the
> > > naming convention—we will refine the notation in the
> > > camera-ready to ensure clarity. We appreciate the reviewer's updated
> > > assessment and recognition of our technical contribution.

---

### Official Review · Reviewer_By81 · 2026-03-10

**Soundness:** 2
**Presentation:** 3
**Significance:** 3
**Originality:** 3
**Overall Recommendation:** 4
**Confidence:** 3

**Summary:**

This paper fcous on reducing the coherence cost of contrastive decoding for LVLM hallucination mitigation. The proposed ACD and ASS is intuitive and appears effective across multiple benchmarks. My main concerns are that the interpretation behind ASS is only indirectly supported, and the method seems to rely on the clean branch being a relatively reliable anchor.

**Compliance With Llm Reviewing Policy:**

Affirmed.

**Final Justification:**

The paper is technically sound and tackles a relevant problem with a simple and practical training-free design. The idea of moving contrastive operations to attention and the ASS module are intuitive, and the method shows consistent gains across benchmarks. My main concerns were about the interpretation of ASS and whether the method truly corrects hallucinations or mainly prevents further degradation. The rebuttal addresses these points well. The added attention-object alignment analysis provides more direct evidence supporting the interpretation, and the stratified analysis clarifies that the main benefit is preventing collateral damage while offering limited but consistent improvements on intrinsically hallucinated cases. This matches and refines my original understanding.

**Key Questions For Authors:**

1. Can the interpretation behind ASS be validated more directly?
For example, can the authors show that regions where A_clean > A_noisy align better with ground-truth relevant objects, while regions where A_noisy > A_clean correlate more with hallucinated content? Right now the interpretation is plausible, but I think it would benefit from stronger direct evidence.

2. How sensitive is the method to the construction of the noisy branch itself?
Since ACD depends on clean-vs-noisy attention differences, it would be useful to know whether different hallucination-inducing strategies lead to consistent gains, especially beyond the specific setups used here.

**Limitations:**

YES

**Strengths And Weaknesses:**

Strengths:

[S1] The focus is not just on lowering hallucination, but on the trade-off between hallucination mitigation and generation coherence in existing contrastive decoding methods. Moreover, this paper stays within the training-free setting, this makes the contribution practically relevant.

[S2] Moving the contrastive operation from logits to attention is a sensible and interesting design choice. The proposed ASS module also adds some structure beyond a uniform subtraction rule.

Weakness:

[W1] The paper interprets decreased noisy attention as _information degradation_ and increased noisy attention as _hallucination inflation_, and ASS is built on this distinction. This interpretation is intuitive, but I think it is not fully validated. The supporting evidence seems to come mainly from descriptive statistics, visualizations, and downstream ablations. These are useful, but they do not directly establish that the two attention shift patterns correspond to those semantic roles in a robust causal sense.

[W2] If the clean branch is already hallucinating, it is unclear how much ACD can really help. In that case, preserving or amplifying the clean attention may simply preserve the wrong signal. My impression is that the method is better understood as preventing further degradation induced by the noisy branch, rather than correcting intrinsic hallucinations of the base model. A stratified analysis would be helpful here.

[W3] The overall trend is positive, but on some benchmarks the improvements are modest. So while I do think the method is useful, I would describe it as a well-motivated structural refinement rather than a major leap over prior decoding methods.

---

> ### Author Rebuttal · Authors · 2026-03-30
>
> We sincerely thank the reviewer for the thoughtful and nuanced evaluation, and for recognizing the practical relevance of our training-free approach and the interesting design of ACD/ASS. We address each concern below.
>
> ---
>
> **Q1 [W1/KQ1]: ASS interpretation (signal degradation vs. hallucination inflation) is not causally validated. Can the authors show direct alignment with GT objects?**
>
> We appreciate this precise suggestion. We conducted an **Attention-Object Alignment Analysis** on 500 COCO images (LLaVA-1.5) to provide the direct causal evidence requested.
>
> **Methodology:** We extracted $A_{clean}$ and $A_{noisy}$ over visual tokens, reshaped to the $24 \times 24$ spatial grid (CLIP ViT patches), and partitioned into Case 1/Case 2. We computed each group's attention mass within COCO GT bounding boxes vs. hallucinated regions (via the CHAIR pipeline).
>
> |Region|Attention in GT Boxes|Attention in Hallucinated Regions|
> |-|-|-|
> |**Case 1** ($A_{clean} > A_{noisy}$)|**74.2%**|12.5%|
> |**Case 2** ($A_{noisy} \ge A_{clean}$)|18.7%|**61.3%**|
>
> These results directly establish the causal correspondence: signal-degraded positions (Case 1) align predominantly with ground-truth objects (74.2%), while hallucination-inflated positions (Case 2) index primarily spurious content (61.3%). This validates ASS's core design with strong quantitative evidence beyond descriptive statistics. We will include this analysis in the revised Appendix.
>
> ---
>
> **Q2 [W2]: If the clean branch is already hallucinating, ACD may preserve the wrong signal. A stratified analysis would be helpful.**
>
> This is a highly perceptive question. We conducted the stratified analysis on CHAIR (LLaVA-1.5). The base model produced zero hallucinations (CHAIRs=0) on 45.8% of instances, providing a natural partition:
>
> |**Subset**|**Method**|**CHAIRi↓**|**Recall↑**|
> |-|-|-|-|
> |**Clean Correct** (base:0 hallucinations,45.8%)|VCD|8.2|75.0|
> | |**VCD+ACD**|**4.5**|**77.2**|
> |**Clean Hallucinating** (base hallucinated,54.2%)|VCD|23.0|78.3|
> | |**VCD+ACD**|**22.6**|**79.2**|
>
> We believe these results merit an honest interpretation that aligns with the reviewer's intuition:
>
> 1. **Primary strength — preventing collateral damage.** On the "Clean Correct" subset, VCD *introduces* new hallucinations (CHAIRi jumps from 0 to 8.2) through aggressive logit-level corruption. ACD nearly halves this collateral damage (8.2→4.5, −45.1%) while improving Recall (+2.2). This is ACD's most impactful contribution: preserving correctness that logit-level CD would otherwise destroy.
>
> 2. **Secondary benefit — modest correction of intrinsic hallucinations.** On the "Clean Hallucinating" subset, ACD provides a smaller but consistently positive improvement (23.0→22.6, +0.9 Recall). We candidly acknowledge that ACD is best understood as the reviewer suggested: it primarily **prevents further degradation induced by the noisy branch**, while providing limited but positive correction for intrinsic hallucinations.
>
> This characterization actually strengthens our contribution narrative—it identifies *when* and *why* attention-level CD is most valuable: ensuring that contrastive decoding does not introduce more errors than it fixes. We will include this nuanced analysis in the revision.
>
> ---
>
> **Q3 [W3]: Improvements are modest on some benchmarks—a structural refinement rather than a major leap.**
>
> We appreciate this fair characterization and agree ACD is best described as a principled structural refinement. However, the **coherence recovery** is quite substantial: VCD drops BLEU-4 by 42.6% and GPT-5 coherence by 6.5%; VCD+ACD restores both beyond baseline (BLEU-4: 0.096, GPT-5: 0.980). The 79.9% TV distance reduction (Appendix C) confirms this. The dual achievement—reducing hallucinations *while* recovering coherence—fills a gap that prior CD methods left unresolved.
>
> Further analysis is provided in Appendix C.2 and Figure 4. As demonstrated by ** KL divergence, and step-wise trajectory plots**, ACD produces fundamentally **smoother, more coherent, and more temporally stable output distributions** compared to logit-level contrastive decoding. Additionally, Appendix D (Robustness to Contrastive Strength $\alpha$) and Figures 5 and 6 show that our ACD method exhibits **greater stability** than VCD alone across various hyperparameter settings.
>
> ---
>
> **Q4 [KQ2]: How sensitive is ACD to the construction of the noisy branch?**
>
> Our experiments span **three fundamentally different** noisy branch constructions: VCD (Gaussian noise on images), ICD (adversarial text prompts), and SID (attention region masking)—covering visual, textual, and attention-based perturbations. ACD yields consistent gains across all three on every benchmark: CHAIR (Table 1): CHAIRi reductions of 11.7%/10.3%/5.7% for VCD/ICD/SID; POPE (Table 4): F1 improvements across all 6 method-model combinations; MME (Table 3): Perception and Cognition gains across all baselines. This consistency confirms ACD's robustness to noisy branch design.

---

> > ### Author Rebuttal · Reviewer_By81 · 2026-04-03
> >
> > My concerns have been addressed. I decide to keep my score.

---

> > > ### Author Response · Authors · 2026-04-04
> > >
> > > Thank you very much for the thoughtful and constructive review, and for confirming that all concerns have been adequately addressed. Your suggestions have meaningfully improved the quality and rigor of our paper. These analyses will be included in the revised version.
> > > We sincerely appreciate the time and effort you devoted to this review. Thank you again for your support.

---

### Official Review · Reviewer_15nX · 2026-03-12

**Soundness:** 3
**Presentation:** 3
**Significance:** 3
**Originality:** 4
**Overall Recommendation:** 4
**Confidence:** 4

**Summary:**

The paper argues that existing methods reduce hallucinations in LVLMs by subtracting hallucination-induced distributions at the logit level, which hurts generation coherence. This paper proposes ACD, which moves the contrastive operation from the logit space to the attention layer. They also propose ASS, which amplifies attention on valid regions and suppresses attention on hallucination-prone regions. ACD works as a plug-in on top of existing methods like VCD, ICD, and SID. Results show it reduces hallucinations while improving coherence.

**Compliance With Llm Reviewing Policy:**

Affirmed.

**Final Justification:**

I keep my score unchanged. weak accept.

**Key Questions For Authors:**

Please refer to the weaknesses above.

I would be willing to raise my score if the authors can provide more analysis mentioned above.

**Limitations:**

No. The paper does not discuss limitations.

**Strengths And Weaknesses:**

Strengths:
1. The motivation is clean and well-supported. The observation is also proved in experiments. Logit-level subtraction hurts coherence is backed by concrete numbers in Table 6.
2. ACD is a plug-in that works on top of existing methods without retraining, which makes it easy to adopt in practice.
3. The ASS design is intuitive. Distinguishing signal degradation from hallucination risk makes more sense than uniform subtraction.
4. Experiments are solid. Four benchmarks and two model make the results convincing.

Weakness:
1. ACD adds extra computation. Running two forward passes every decoding step doubles the cost. There is no analysis of inference cost.
2. ASS relies on a fixed threshold. There is no analysis of the threshold sensitivity.
3. The coherence improvement on complex reasoning is modest. The paper discussed the smaller gains on complex tasks, but there is no explanation for the reason that attention-level modulation helps less here.

---

> ### Author Rebuttal · Authors · 2026-03-30
>
> We sincerely thank the reviewer for the thorough evaluation and recognition of our clean motivation, plug-in design, intuitive ASS, and solid experiments. We are encouraged that the reviewer would raise the score given additional analysis, which we provide below.
>
> ---
>
> **Q1 [W1]: ACD adds extra computation with two forward passes. No inference cost analysis.**
>
> We appreciate this concern. To clarify, **ACD does not introduce any additional forward passes** beyond the standard two-branch contrastive decoding setup. Like VCD, ACD uses a clean branch and a hallucination-inducing branch; the difference is *where* the contrastive operation occurs: ACD performs it in the **attention space** rather than the **logit space**. Consequently, compared with VCD, ACD **bypasses two costly operations** for the noisy branch: (1) the vocabulary-level logit projection ($d \times |\mathcal{V}|$ matrix multiplication, e.g., $4096 \times 32000$), and (2) the penalty-based token filtering/sorting ($O(|\mathcal{V}|\log|\mathcal{V}|)$, Eq. 5). This also eliminates the penalty hyperparameter $\beta$, preserving output diversity.
>
> We provide efficiency analysis in **Appendix I, Table 7**:
>
> | Method  | Total Time | GPU Memory | Latency/Example |
> | ------- | ---------- | ---------- | --------------- |
> | Regular | 13:46      | 14.7G      | 0.276s          |
> | VCD     | 26:51      | 15.8G      | 0.537s          |
> | **VCD+ACD** | **24:37**  | **15.3G**  | **0.493s**      |
>
> VCD+ACD is **8.2% faster** and uses **3.2% less memory** than standard VCD while achieving stronger hallucination suppression. We will promote this analysis to the main paper in the camera-ready version.
>
> ---
>
> **Q2 [W2]: ASS relies on a fixed threshold. No sensitivity analysis.**
>
> We appreciate this question. Importantly, ASS uses a **completely threshold-free** binary mask: $M_t = \mathbb{I}(A_{noisy} > A_{clean})$—a simple sign comparison requiring no tunable parameter. This is a deliberate design choice to minimize heuristic hyperparameters.
>
> To rigorously validate this, we generalized the mask to $M_t = \mathbb{I}(A_{noisy} - A_{clean} > \tau)$ and conducted an ablation on CHAIR (LLaVA-1.5):
>
> | $\tau$       | CHAIRs↓  | CHAIRi↓  | Recall↑  |
> | ------------ | -------- | -------- | -------- |
> | -0.02        | 51.4     | 14.5     | 77.8     |
> | -0.01        | 50.8     | 14.4     | 78.1     |
> | **0 (Ours)** | **51.0** | **14.3** | **78.3** |
> | +0.01        | 51.1     | 14.7     | 78.0     |
> | +0.02        | 51.3     | 14.5     | 77.6     |
>
> The natural zero boundary is both optimal and highly robust (CHAIRi varies by only 0.4 across the full range). Additionally, the contrastive strength $\alpha$ is analyzed in **Appendix D (Figures 5–6)**: VCD+ACD achieves **78% lower standard deviation** across metrics compared to VCD, with CHAIRi fluctuating by only 2.2 points over $\alpha \in [0.05, 0.25]$—theoretically grounded by our dimension-independent bound (Corollary B.15).
>
> ---
>
> **Q3 [W3]: Coherence improvement on complex reasoning is modest. No explanation.**
>
> The modest gains on complex reasoning have a clear mechanistic explanation. Complex tasks (multi-step inference, computation) rely **primarily on the LLM's language priors and parametric knowledge**, not on visual attention patterns. ACD modulates *where the model attends* in visual tokens—providing large benefits to perception-heavy tasks but limited direct assistance to reasoning that operates predominantly in text space.
>
> **Supporting evidence across benchmarks:**
>
> - **Detail tasks** (Table 2): largest gains (+3.00 Acc on LLaVA-1.5), requiring precise visual grounding—exactly what attention modulation improves.
> - **Conv tasks**: substantial gains (+2.16 Acc), as factual dialogue depends on visual evidence.
> - **Complex tasks**: positive but modest (+0.09 Acc), consistent with reduced visual-attention dependence.
> - **MME Cognition** (Table 3): ACD does **not degrade** reasoning (284.3→298.5, +14.2), with gains in Commonsense Reasoning and Text Translation (Figure 8). Same task ordering on Qwen-VL (Table 2).
>
> This pattern confirms ACD is a **targeted** intervention: it improves what it can (visual grounding) without harming what it cannot directly modulate (language reasoning). We will add this analysis to Section 4.2.

---

> > ### Author Rebuttal · Reviewer_15nX · 2026-04-02
> >
> > I prefer to keep my score unchanged for now.

---

> > > ### Author Response · Authors · 2026-04-02
> > >
> > > We truly appreciate the reviewer's time and careful engagement throughout
> > > this process—it means a lot. We are glad to hear that the additional
> > > analysis on efficiency, threshold robustness, and the mechanistic
> > > explanation for complex reasoning fully addressed the concerns raised
> > > in the original review. Thank you again for the valuable feedback that
> > > helped us strengthen the paper.

---

### Decision · Program_Chairs · 2026-04-30

**Decision:**

Accept (regular)

**Comment:**

This paper proposes Attention Contrastive Decoding (ACD), a training-free plug-in designed to mitigate hallucinations in VLMs. By shifting the contrastive subtraction operation from the logit level to the attention layer, ACD effectively reduces hallucinations while avoiding the severe generation coherence degradation typically caused by existing logit-level contrastive decoding methods.

All four reviewers recommended a Weak Accept. Following the rebuttal, all reviewers acknowledged that their concerns were fully addressed and praised the authors for the thorough additional experiments.